# A hot-Jupiter progenitor on a super-eccentric retrograde orbit

Arvind F. Gupta[1,2,3]✉, Sarah C. Millholland[4,5], Haedam Im[4,5], Jiayin Dong[6], Jonathan M. Jackson[7], Ilaria Carleo[8,9], Jessica Libby-Roberts[2,3], Megan Delamer[2,3], Mark R. Giovinazzi[10], Andrea S. J. Lin[2,3], Shubham Kanodia[11], Xian-Yu Wang[12], Keivan Stassun[13], Thomas Masseron[8,9], Diana Dragomir[14], Suvrath Mahadevan[2,3], Jason Wright[2,3,15], Jaime A. Alvarado-Montes[16,17], Chad Bender[18], Cullen H. Blake[10], Douglas Caldwell[19], Caleb I. Cañas[20], William D. Cochran[21,22], Paul Dalba[23], Mark E. Everett[1], Pipa Fernandez[1], Eli Golub[1], Bruno Guillet[38], Samuel Halverson[24], Leslie Hebb[25,26], Jesus Higuera[1], Chelsea X. Huang[27], Jessica Klusmeyer[1], Rachel Knight[38], Liouba Leroux[38], Sarah E. Logsdon[1], Margaret Loose[38], Michael W. McElwain[20], Andrew Monson[18], Joe P. Ninan[28], Grzegorz Nowak[8,9,29], Enric Palle[8,9], Yatrik Patel[1], Joshua Pepper[30], Michael Primm[38], Jayadev Rajagopal[1], Paul Robertson[31], Arpita Roy[32], Donald P. Schneider[2,3], Christian Schwab[16,17], Heidi Schweiker[1], Lauren Sgro[19], Masao Shimizu[38], Georges Simard[38], Guðmundur Stefánsson[33,34], Daniel J. Stevens[35], Steven Villanueva[20], John Wisniewski[36], Stefan Will[38] & Carl Ziegler[37]

Giant exoplanets orbiting close to their host stars are unlikely to have formed in their present configurations[1]. These 'hot Jupiter' planets are instead thought to have migrated inward from beyond the ice line and several viable migration channels have been proposed, including eccentricity excitation through angular-momentum exchange with a third body followed by tidally driven orbital circularization[2,3]. The discovery of the extremely eccentric ($e = 0.93$) giant exoplanet HD 80606 b (ref. 4) provided observational evidence that hot Jupiters may have formed through this high-eccentricity tidal-migration pathway[5]. However, no similar hot-Jupiter progenitors have been found and simulations predict that one factor affecting the efficacy of this mechanism is exoplanet mass, as low-mass planets are more likely to be tidally disrupted during periastron passage[6–8]. Here we present spectroscopic and photometric observations of TIC 241249530 b, a high-mass, transiting warm Jupiter with an extreme orbital eccentricity of $e = 0.94$. The orbit of TIC 241249530 b is consistent with a history of eccentricity oscillations and a future tidal circularization trajectory. Our analysis of the mass and eccentricity distributions of the transiting-warm-Jupiter population further reveals a correlation between high mass and high eccentricity.

The Transiting Exoplanet Survey Satellite (TESS)[9] monitored the apparent brightness of the star TIC 241249530 for 28 days during the second year of its primary mission. These data reveal a transit-like approximately 0.8% dip in brightness on 12 January 2020, the shape and depth of which were consistent with a Jupiter-sized planet passing in front of the star (Fig. 1a). To find out the nature and origin of this signal, we conducted a series of ground-based observations of TIC 241249530. We first used high-spatial-resolution speckle imaging data from NESSI[10] to rule out the presence of contaminating sources and confirm that the signal was not associated with a background eclipsing-binary in the TESS aperture. We then began radial velocity (RV) observations with the NEID spectrograph[11], which revealed that the TESS transit was probably induced by a giant exoplanet on a highly eccentric ($e = 0.94$), long-period ($P = 167$ days) orbit. These measurements were consistent with the absence of a

transit detection when TESS re-observed this star for 27 days from December 2022 to January 2023. Further NEID measurements, supplemented by observations with the HPF[12] and HARPS-N[13] spectrographs, were strategically scheduled to be taken when the planet was predicted to be approaching periastron and thus inducing large stellar RV variations. We attempted to detect a second transit using the global Unistellar telescope network[14] in March 2023, but these efforts were unsuccessful as the ephemeris was not yet well constrained. However, RV data collected during the periastron window enabled us to more precisely predict the subsequent transit window. We captured the first half of this transit using the engineered diffuser[15] on the ARCTIC imager[16] on 30 August 2023 (Fig. 1b). We refined the orbit using the ARCTIC data together with further NEID observations, including several concurrent with this transit and the subsequent one on 12 February 2024. Our ensemble of photometric

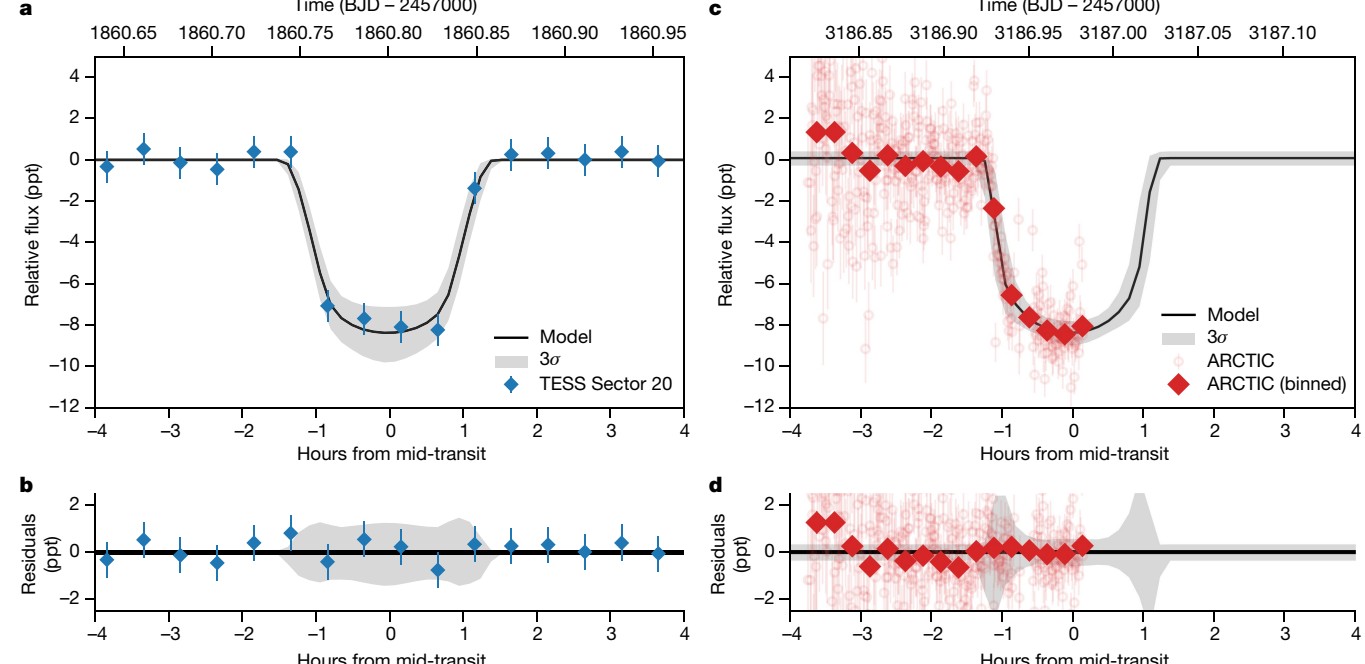

**Fig. 1 | TIC 241249530 b transit measurements. a**, TESS photometric measurements (blue), shown at their native 30-min cadence, and the best-fit transit model (black curve) and 3σ confidence region (grey). **b**, Residuals to the best-fit model for the TESS transit detection. **c**, Diffuser-assisted ARCTIC photometric measurements (red) and the best-fit transit model (black curve) and 3σ confidence region (grey). We show both the raw 30-s cadence and binned 15-min cadence measurements, along with the residuals to the fitted transit signal. **d**, Residuals to the best-fit model for the ARCTIC transit detection. All brightnesses are given in parts per thousand (ppt). Error bars on individual data points indicate the 1σ measurement uncertainties.

and RV measurements is best explained by a massive exoplanet on a long-period, eccentric orbit.

To characterize the host star TIC 241249530, we separately analysed the NEID and HARPS-N spectra using synthetic spectral fitting techniques and we then fit the spectral energy distribution (SED; see Methods). TIC 241249530 is a main-sequence star that is slightly hotter, larger and more massive than the Sun; the derived parameters, listed in Extended Data Table 1, suggest that the star is 3.2 ± 0.5 Gyr old. The star also has a low-mass binary stellar companion, TIC 241249532, at a projected separation of 4.930 ± 0.104″, or 1,664 ± 11 AU.

We jointly fit the NEID, HPF, HARPS-N, TESS and ARCTIC measurements, accounting for perturbations to the in-transit RV signal owing to the Rossiter–McLaughlin effect. The transit and RV fits are shown in Figs. 1 and 2, respectively, and the best-fit parameters are given in Table 1. TIC 241249530 b is an exoplanet that is $4.98^{+0.16}_{-0.18}$ times as massive as Jupiter and it is on a $165.77190^{+0.00027}_{-0.00028}$-day orbit around its host star, with an eccentricity of $0.9412^{+0.0009}_{-0.0009}$. Our fit to the Rossiter–McLaughlin signal (Fig. 2b) shows that the exoplanet is orbiting in the opposite direction to the projected stellar spin ($\lambda = 163.5^{+9.4°}_{-7.7}$) and is retrograde to 99.5% confidence. Few exoplanets have orbits as extreme as this; this orbit is more eccentric than that of any other transiting exoplanet, and only a handful of known planets have similarly large projected spin–orbit misalignments[17].

The planet that most closely resembles TIC 241249530 b is HD 80606 b (ref. 4), which has a mass 4.1 times that of Jupiter and is also on a misaligned orbit with a period of 111 days and an eccentricity of 0.93. HD 80606 b is an archetypal example of an exoplanet destined to become a hot Jupiter with an eventual orbital period of less than 10 days. The eccentric orbit carries the planet close enough to its host star at periastron that tides raised on the planet and star will sap energy from the orbit, causing it to gradually shrink and circularize. Also, simulations[3,5] show that the present orbit of HD 80606 b is consistent with a

history of von Zeipel–Lidov–Kozai (vZLK) eccentricity oscillations[18–20] driven by angular momentum exchange with HD 80607, the stellar companion to the host star. Our own simulations of the dynamical history and trajectory of TIC 241249530 b (see Methods) show that the orbit is consistent with this same type of perturber-coupled, high-eccentricity tidal migration. Eccentricity oscillations would have continued until the most recent few hundred million years, at which point general relativistic precession overtook the torque exerted by the companion, locking

**Table 1 | TIC 241249530 b system parameters**

| Parameter | Value | Description |
|---|---|---|
| $T_0$ | $2458860.8007^{+0.0015}_{-0.0016}$ | Time of mid-transit (BJD) |
| $P$ | $165.77190^{+0.00027}_{-0.00028}$ | Orbital period (days) |
| $e$ | $0.9412^{+0.0009}_{-0.0009}$ | Orbital eccentricity |
| $\omega$ | $42.32^{+0.40}_{-0.36}$ | Argument of periastron (°) |
| $i$ | $85.17^{+0.57}_{-0.51}$ | Orbital inclination (°) |
| $K$ | $463.3^{+4.1}_{-4.0}$ | RV semi-amplitude (m s$^{-1}$) |
| $M_p$ | $4.98^{+0.16}_{-0.18}$ | Exoplanet mass ($M_J$) |
| $R_p$ | $1.19^{+0.04}_{-0.04}$ | Exoplanet radius ($R_J$) |
| $\lambda$ | $163.5^{+9.4}_{-7.7}$ | Projected spin–orbit obliquity (°) |
| $M_\star$ | $1.271^{+0.061}_{-0.068}$ | Stellar mass ($M_\odot$) |
| $R_\star$ | $1.397^{+0.025}_{-0.028}$ | Stellar radius ($R_\odot$) |
| $v\sin i_\star$ | $4.60^{+0.56}_{-0.63}$ | Projected rotational velocity (km s$^{-1}$) |

We report the median values of the posterior distributions from our joint fit to the observed transits and RVs. The uncertainties represent the 68% confidence intervals (±1σ) for each parameter.

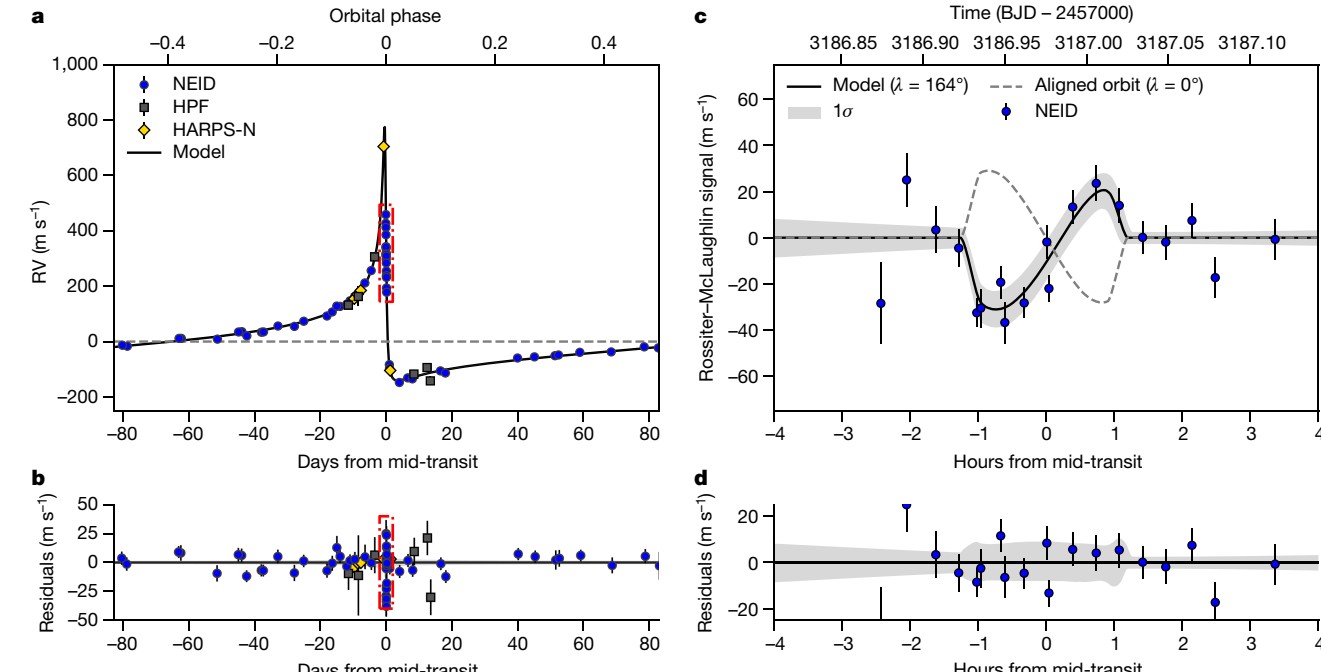

**Fig. 2 | Phase-folded RV measurements for TIC 241249530. a**, RV measurements from NEID (blue), HPF (black), HARPS-N (yellow) and best-fit orbit model (black curve). **b**, Residuals to the RV orbit fit. **c**, Best-fit Rossiter–McLaughlin model (solid curve) and $1\sigma$ confidence region (grey), with the signal for aligned orbit shown for comparison (dashed curve). **d**, Residuals to the Rossiter–McLaughlin model fit. The dashed red box in **a** highlights the in-transit RV measurements, which are shown in **b**. Error bars on individual data points indicate the $1\sigma$ measurement uncertainties.

the exoplanet on an eccentric orbit that is now gradually circularizing. The architectures of the HD 80606 and TIC 241249530 systems lend support to this process as a plausible hot-Jupiter-formation mechanism. However, although other giant exoplanets on tidal migration tracks have been discovered[21], including two that probably have vZLK-driven dynamical histories[22,23], no previous examples have eccentricities >0.9 and none have formation scenarios as clear as that of HD 80606 b. The observed occurrence rate of super-eccentric progenitors to hot Jupiters[24,25] falls well short of predictions from simulations[26], suggesting that giant-planet migration is dominated by other channels. With the discovery of TIC 241249530 b, a second super-eccentric exoplanet in a hierarchical triple system has been added to the sparse sample, providing a new lens through which we can explore the formation of these planets.

Not only do the TIC 241249530 b and HD 80606 b systems share similar orbital architectures but these exoplanets also have similar masses. The masses and eccentricities of all transiting warm Jupiters, which we define as giant planets with intermediate periods (10 days < $P$ < 365 days), are shown in Fig. 3b. These two planets, which are the only members of the sample with super-eccentric orbits ($e$ > 0.9), are also among the most massive. A correlation between exoplanet mass and eccentricity has been identified in previous works[27–30], each of which found that higher-mass planets are more likely to have larger orbital eccentricities. We find that our narrower sample of transiting giant planets conforms to this known trend (see Methods); the eccentricity distributions of high-mass ($M_p$ > 2 $M_J$) and low-mass (0.3 $M_J$ ≤ $M_p$ ≤ 2 $M_J$) members of this population are statistically distinct (Fig. 3a). Although lower-mass planets are more likely to be found on low-eccentricity orbits, high-mass planets exhibit a broad, nearly flat distribution from circular to highly eccentric orbits.

Although the observed mass–eccentricity correlation may be shaped by several processes, such as collisional eccentricity growth[30] or resonant interactions with the protoplanetary disk[31–33], the high masses

of TIC 241249530 b and HD 80606 b may offer a clue as to the dearth of super-eccentric giant planets. During the high-eccentricity phase of vZLK oscillations, orbital eccentricities can be driven so close to unity that the exoplanet will approach, or even breach, the tidal radius of the host star. Because the tidal radius is inversely proportional to the planet–star mass ratio, lower-mass planets more easily cross this threshold and experience tidal disruption. A relative dearth of low-mass, eccentric progenitors to hot Jupiters is a consistent outcome of simulations of high-eccentricity tidal migration following vZLK oscillations under an equilibrium tide assumption[6–8]. For planets susceptible to chaotic, or diffusive, dynamical tidal evolution, whereby oscillations excited in the planet accelerate orbital decay, this mass dependence is largely erased, as low-mass planets can become decoupled from the perturber before being disrupted[34]. However, chaotic tides facilitate circularization on a much shorter timescale (<100 Myr); these planets will spend very little time with intermediate-period orbits[34–36]. It is possible that only the most massive eccentric giant planets last long enough in this period regime to be represented in the observed sample.

TIC 241249530 b passes through periastron just six hours before each transit, presenting a unique opportunity to observe how an exoplanet atmosphere responds to a rapid, extreme heating event. Temporal variations in exoplanet atmospheres are best explored through studies of planets on eccentric orbits, for which we may see signatures of time-varying irradiation and changing pressure–temperature profiles, such as turbulent surface flows[37] and disequilibrium chemistry[38], depending on the heat-redistribution timescales. The atmospheres of several eccentric giant planets have been studied[39–41], but the periastron phase has not been captured in transit for these systems. The orbital geometry of TIC 241249530 b will make such measurements possible for the first time. The planetary atmosphere can also be studied by means of emission measurements during other orbital phases, but the orientation precludes a secondary eclipse at a 6.3$\sigma$ confidence.

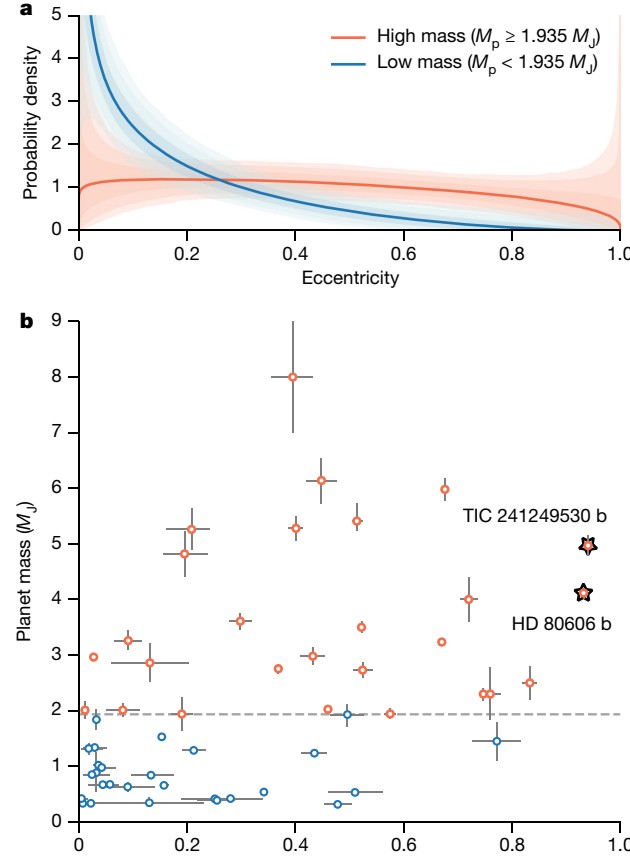

**Fig. 3 | Mass–eccentricity distribution for transiting warm Jupiters.**
**a**, Best-fit beta distributions for transiting warm Jupiters more massive (red) and less massive (blue) than 1.935 $M_J$. The shaded regions represent the 1$\sigma$ (darkest), 2$\sigma$ and 3$\sigma$ (lightest) posteriors on each fit. **b**, Masses and eccentricities of transiting warm Jupiters. The super-eccentric hot-Jupiter progenitors HD 80606 b and TIC 241249530 b are labelled with upright and inverted stars, respectively. The horizontal dashed line indicates the median mass of the population, which is the threshold used for the fits shown in **a**. Our results are insensitive to the exact value of the threshold (see Methods); the median is chosen simply for visualization purposes. Error bars on individual data points indicate the 1$\sigma$ uncertainties from the literature.

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

¹U.S. National Science Foundation National Optical-Infrared Astronomy Research Laboratory (NSF NOIRLab), Tucson, AZ, USA. ²Department of Astronomy and Astrophysics, The Pennsylvania State University, University Park, PA, USA. ³Center for Exoplanets and Habitable Worlds, The Pennsylvania State University, University Park, PA, USA. ⁴Department of Physics,

Massachusetts Institute of Technology, Cambridge, MA, USA. [5]Kavli Institute for Astrophysics and Space Research, Massachusetts Institute of Technology, Cambridge, MA, USA. [6]Center for Computational Astrophysics, Flatiron Institute, New York, NY, USA. [7]Van Vleck Observatory, Astronomy Department, Wesleyan University, Middletown, CT, USA. [8]Instituto de Astrofísica de Canarias (IAC), La Laguna, Tenerife, Spain. [9]Departamento de Astrofísica, Universidad de La Laguna (ULL), La Laguna, Tenerife, Spain. [10]Department of Physics and Astronomy, University of Pennsylvania, Philadelphia, PA, USA. [11]Earth and Planets Laboratory, Carnegie Institution for Science, Washington DC, USA. [12]Department of Astronomy, Indiana University Bloomington, Bloomington, IN, USA. [13]Department of Physics and Astronomy, Vanderbilt University, Nashville, TN, USA. [14]Department of Physics and Astronomy, University of New Mexico, Albuquerque, NM, USA. [15]Penn State Extraterrestrial Intelligence Center, The Pennsylvania State University, University Park, PA, USA. [16]School of Mathematical and Physical Sciences, Macquarie University, North Ryde, New South Wales, Australia. [17]The Macquarie University Astrophysics and Space Technologies Research Centre, Macquarie University, North Ryde, New South Wales, Australia. [18]Department of Astronomy and Steward Observatory, University of Arizona, Tucson, AZ, USA. [19]Carl Sagan Center, SETI Institute, Mountain View, CA, USA. [20]NASA Goddard Space Flight Center, Greenbelt, MD, USA. [21]Center for Planetary Systems Habitability, The University of Texas at Austin, Austin, TX, USA. [22]McDonald Observatory, The University of Texas at Austin, Austin, TX, USA. [23]Department of Astronomy and Astrophysics, University of California, Santa Cruz, Santa Cruz, CA, USA. [24]Jet Propulsion Laboratory, California Institute of Technology, Pasadena, CA, USA. [25]Physics Department, Hobart and William Smith Colleges, Geneva, NY, USA. [26]Department of Astronomy, Cornell University, Ithaca, NY, USA. [27]Centre for Astrophysics, University of Southern Queensland, Toowoomba, Queensland, Australia. [28]Department of Astronomy and Astrophysics, Tata Institute of Fundamental Research, Mumbai, India. [29]Institute of Astronomy, Faculty of Physics, Astronomy and Informatics, Nicolaus Copernicus University, Toruń, Poland. [30]Department of Physics, Lehigh University, Bethlehem, PA, USA. [31]Department of Physics & Astronomy, University of California, Irvine, Irvine, CA, USA. [32]Schmidt Sciences, New York, NY, USA. [33]Department of Astrophysical Sciences, Princeton University, Princeton, NJ, USA. [34]Anton Pannekoek Institute for Astronomy, University of Amsterdam, Amsterdam, The Netherlands. [35]Department of Physics & Astronomy, University of Minnesota Duluth, Duluth, MN, USA. [36]NASA Headquarters, Washington DC, USA, . [37]Department of Physics, Engineering & Astronomy, Stephen F. Austin State University, Nacogdoches, TX, USA. [38]Unaffiliated: Bruno Guillet, Rachel Knight, Liouba Leroux, Margaret Loose, Michael Primm, Masao Shimizu, Georges Simard, Stefan Will. ✉e-mail: arvind.gupta@noirlab.edu

## Methods

### TESS photometry

TIC 241249530 was observed with TESS[9] from 24 December 2019 to 21 January 2020 (Sector 20) at a 30-min cadence and from 21 December 2022 to 18 January 2023 (Sector 60) at a 2-min cadence. A single transit-like dip (flux depth about 8 parts per thousand) was identified by the TESS Single Transit Planet Candidate Working Group (TSTPC WG) in the Sector 20 Quick Look Pipeline[42,43] light curve using a box least-squares search. The TSTPC WG focuses on searching full-frame TESS light curves for isolated transit events and validating and confirming those that are true planets, with the aim of increasing the yield of TESS planets with period >30 days (for example, refs. 44–47). There is no flux centroid motion during the transit event for TIC 241249530 and we identify no other sources brighter than $\Delta m_{\rm G} = 5$ in the target aperture. Although there is flux contamination from two nearby stars with $6 > \Delta m_{\rm G} > 5$, TIC 241249532 and TIC 241249533, both of which were centred on the same pixel as TIC 241249530 in Sector 20, these are too faint to have been responsible for the observed change in brightness. No notable brightness variations were detected in the Sector 60 light curve. For all subsequent analysis in this work, we rely on the pre-search data conditioned simple aperture photometry[48–50] (PDCSAP) light curve from the Science Processing Operations Center[51] (SPOC) for Sector 60 and the TESS-SPOC[52] light curve for Sector 20 (Extended Data Fig. 1).

### High-contrast imaging

To verify that the transit signature detected by TESS was indeed associated with TIC 241249530 and not with a nearby star or binary system that was blended in the TESS aperture, we used the NN-EXPLORE Exoplanet Stellar Speckle Imager (NESSI)[10] on the WIYN 3.5-m telescope at Kitt Peak National Observatory to conduct high-spatial-resolution observations of the target on 21 April 2021. A sequence of 1,000 40-ms exposures was taken in the 832-nm and 562-nm narrow-band filters simultaneously with the red and blue NESSI cameras, respectively. These diffraction-limited exposures were used to reconstruct high-contrast images (Extended Data Fig. 2) following the steps outlined in ref. 53. The achieved $5\sigma$ contrast limits are sufficient to rule out the presence of faint stellar companions and background sources with $\Delta {\rm mag}_{562} < 3.3$ and $\Delta {\rm mag}_{832} < 3.7$ at a separation of 0.2″ and $\Delta {\rm mag}_{562} < 3.9$ and $\Delta {\rm mag}_{832} < 4.8$ at a separation of 1″.

### Ground-based photometric observations

We used the Unistellar Network, a collaboration of citizen scientists using Unistellar telescopes[54] in support of astronomical research, to observe TIC 241249530 from locations in Japan, Europe and the United States in search for transit signatures in March 2023. Observations were taken at various times from 7 to 19 March 2023, when the companion orbital period and transit ephemeris were still highly uncertain. After removing off-target and saturated frames, we calibrated the remaining images, binned them in sets of 15–30 to amplify the signal-to-noise ratio (S/N) and performed differential photometry[55,56]. No signatures of statistical significance were found in the Unistellar data and, based on our subsequent orbit fit, we confirm that none of these observations were taken during the transit.

We observed TIC 241249530 again on 30 August 2023 with the Astrophysical Research Consortium Telescope Imaging Camera (ARCTIC)[16] on the ARC 3.5-m telescope at Apache Point Observatory (APO). Observations were conducted using a beam-shaping diffuser, which creates a stable top-hat point spread function of the star to improve photometric precision[15]. We used the Semrock narrow-band filter (838–876 nm) to avoid atmospheric absorption bands[57]. We began observing when the target rose above an air mass of 4 (altitude ≈ 10°) and continued until 12° morning twilight, collecting a continuous 4.3-h baseline of consecutive 30-s exposures. As the star rose above air mass approximately 1.5, about

2.5 h after the start of the observing sequence, a transit-like decrease in brightness was observed.

To reduce the ARCTIC data, a median-combined master bias image was constructed and subtracted from the individual science frames, which were flat-fielded using dome flat exposures taken at the start of the night. We performed differential aperture photometry on the reduced data using AstroImageJ[58] with a 17-pixel (7.7″) aperture and four reference stars that were carefully selected to minimize the scatter of the out-of-transit flux. Flux uncertainties were calculated following the procedures in refs. 15,59, which account for photon noise from the star and background, detector read noise and air-mass-dependent scintillation noise. We removed exposures flagged by AstroImageJ for approaching the detector saturation limit, as well as exposures taken during intermittent cloud cover that introduced further scatter.

The diffused point spread function of TIC 241249530 overlapped with that of TIC 241249532. Before initiating our ARCTIC observing sequence, we collected several individual exposures without the diffuser in the optical path. We used these data to calculate the relative brightness contributions of the two stars. TIC 241249532 contributes just 0.53% of the total flux in the Semrock bandpass.

### Spectroscopic observations

We monitored the RV signal of TIC 241249530 with the NEID spectrograph[11] on the WIYN 3.5-m telescope, collecting measurements on 40 separate nights between 2 September 2021 and 1 March 2024. On all but three of these nights, single exposures were taken, with exposure times ranging from 500 to 1,800 s, depending on the observing conditions. On the night of 30 August 2023, four consecutive 20-min exposures were taken simultaneously with the partial transit as observed with ARCTIC, and on the subsequent night, we secured a pair of measurements separated by an hour. We also obtained a sequence of 15 consecutive 20-min exposures on the night of 12 February 2024; this sequence covered a full transit as well as several measurements before ingress and after egress. We discard two spectra that were taken on nights for which the wavelength calibration was identified to be unreliable, leaving us with 56 high-quality measurements with a median S/N per extracted pixel of 25 at 550 nm. The raw echelle spectra were processed with version 1.3 of the NEID Data Reduction Pipeline (DRP; https://neid.ipac.caltech.edu/docs/NEID-DRP/), which produces wavelength-calibrated 1D spectra and then calculates RVs using the cross-correlation function (CCF) method[60]. We also independently calculated the RVs from the calibrated 1D spectra using a modified version of the SpEctrum Radial Velocity AnaLyser (SERVAL) template-matching algorithm[61,62] that has been optimized for NEID spectra as described by ref. 63. The SERVAL RVs were calculated using the central 7,000 pixels of 79 orders centred between 4,010 and 8,400 Å (order indices 20 to 100, corresponding to echelle orders 153 to 73). The template-matching results outperform the CCF-based RVs from the NEID DRP, with median single measurement precisions of $\sigma_{\rm RV,SERVAL} = 6.3$ m s$^{-1}$ and $\sigma_{\rm RV,DRP} = 7.9$ m s$^{-1}$, so we chose to use the SERVAL RVs for the analysis performed in this work.

Further RV measurements were taken with the Habitable-zone Planet Finder (HPF) spectrograph[12], which is on the Hobby–Eberly Telescope (HET)[64,65] at McDonald Observatory, and the HARPS-N spectrograph, mounted on Telescopio Nazionale Galileo (TNG) in La Palma, as TIC 241249530 approached periastron in March 2023. Six HPF observations were made between 6 and 31 March 2023, for which each observation consisted of two consecutive 945-s exposures with a median nightly binned S/N per extracted pixel of 137 at 1,000 nm. These data were processed using the HxRGproc[66] and barycorrpy[67] packages and the RVs were calculated using a version of SERVAL that has been modified for HPF[68,69]. We achieve a median RV measurement precision of 15.0 m s$^{-1}$. We also observed the target five times with HARPS-N between 7 and 18 March 2023, with an exposure time of 3,300 s and a mean (min, max) S/N of 55 (37, 75). We reduced the data with the offline version of the HARPS-N data-reduction software through the Yabi web

interface[70] installed at the Italian Center for Astronomical Archives Data Center. To extract the RVs, we used a G2 mask template and obtained a CCF width of 9.9 km s$^{-1}$, with an average precision of 0.1 km s$^{-1}$. The median resulting RV measurement precision is 3.4 m s$^{-1}$. We show the complete RV time series from NEID, HPF and HARPS-N in Extended Data Fig. 3.

## Stellar characterization of TIC 241249530

To determine the stellar atmospheric parameters of TIC 241249530, we analysed the out-of-transit NEID spectra collected before September 2023 (cumulative S/N ≈ 100 at 550 nm) using the iSpec[71,72] Python package to perform synthetic spectral fitting. We used the SPECTRUM radiative transfer code[73], MARCS atmospheric models[74], solar abundances from 3D hydrodynamic models[75] and the sixth version of the Gaia ESO survey (GES) atomic line list[76]. The microturbulence velocity was treated as a free parameter to allow for flexibility in accounting for small-scale motions in the stellar atmosphere. Macroturbulence was determined using an empirical relation, making use of established correlations with other stellar properties[77]. To streamline the fitting, we restricted the analysis to specific spectral regions from 480 to 680 nm, encompassing the wing segments of the Hα, Hβ and Mg I triplet lines, which are sensitive to $T_{eff}$ and log$g$, and the Fe I and Fe II lines, which provide precise constraints on [Fe/H] and $v$sin$i_\star$. We minimize the difference between the synthetic and input spectra by applying the nonlinear least-squares Levenberg–Marquardt fitting algorithm, using constraints from the aforementioned models and line lists.

The HARPS-N spectra were independently analysed with BACCHUS[78], using MARCS atmospheric models, the GES atomic line list and the TUR-BOSPECTRUM radiative transfer code[79,80]. For our fit, we constrained $T_{eff}$ by requiring Fe I line abundances to be uncorrelated with their respective excitation potentials in the synthetic spectrum and we constrained log$g$ by requiring ionization balance for the Fe I and Fe II lines. We also required the Fe I line abundances to be uncorrelated with their equivalent widths and the stellar metallicity ([Fe/H]) was calculated as the average of these abundances. The projected rotational velocity was estimated by fitting the broadening of the Fe I lines, accounting for the best-fit microturbulence and assuming the same macroturbulence contribution as in the iSpec analysis. The stellar parameters derived from the NEID and HARPS-N spectra are largely in good agreement (<1$\sigma$). Discrepancies between the [Fe/H] values and $v$sin$i_\star$ values at the 1.2$\sigma$ level probably result from differences between the fitted microturbulence, which is known to exhibit small variations for different fitting methods[72]. We adopt the iSpec $T_{eff}$, log$g$, [Fe/H] and $v$sin$i_\star$ for the rest of the analysis in this work.

We performed an analysis of the broadband SED of TIC 241249530 together with the Gaia DR3 parallax following the procedures described in refs. 81–83. We use $JHK_S$ magnitudes from 2MASS[84], W1–W3 magnitudes from WISE[85], $G_{BP}G_{RP}$ magnitudes from Gaia[86], BVgri magnitudes from APASS[87] and the NUV magnitude from GALEX[88]. We also used the Gaia spectrophotometry spanning 0.4–1.0 μm. Altogether, the available photometry spans the full stellar SED over the wavelength range 0.2–10.0 μm. We fit the SED using PHOENIX stellar atmosphere models[89], with the effective temperature, surface gravity and metallicity set to the spectroscopically determined values. The remaining free parameter is the extinction ($A_V$), which we limited to the maximum line-of-sight value of $A_V$ = 0.44 mag from galactic dust maps[90]. The resulting fit is shown in Extended Data Fig. 4. Integrating the unreddened model SED yields the bolometric flux at Earth, $F_{bol}$ = 7.19 ± 0.20 × 10$^{-10}$ erg s$^{-1}$ cm$^{-2}$. Taking the $F_{bol}$ and $T_{eff}$ together with the Gaia parallax, we calculate the stellar radius to be $R_\star$ = 1.404 ± 0.028 $R_\odot$. Also, the stellar mass is inferred using empirical relations[91], giving $M_\star$ = 1.24 ± 0.07 $M_\odot$, and we estimate the age to be 3.2 ± 0.5 Gyr by fitting the evolutionary state with the Yonsei–Yale isochrone models[92]. Our reported 0.5-Gyr uncertainty accounts for the uncertainties on each of the inputs to the isochrone fit: effective temperature, surface gravity, metallicity and stellar mass. However, this does not account for systematic uncertainties arising from our choice of stellar models, which can be on the order of 1 Gyr.

The best-fit extinction for our SED model is $A_V$ = 0.31 ± 0.02. This large value is supported by a clear detection of interstellar absorption in the Na D doublet and the K I 770 nm lines in the NEID spectra. Both spectroscopic analyses yield $T_{eff}$ values that are substantially hotter than the literature value from Gaia DR3 spectrophotometric analysis[92], which is consistent with the effect of reddening from dust along the line of sight to the star.

Using the projected rotational velocity and stellar radius, we place an upper limit on the rotation period of $16.9^{+3.8}_{-2.6}$ days. We attempt to make a more precise measurement of the rotation period to determine the stellar inclination, but the existing data are insufficient. An analysis of the TESS light curves using the TESS Systematics-Insensitive Periodogram package[93] shows no notable photometric modulation on timescales shorter than the length of each individual sector. We also examine archival photometry of the star from the WASP survey[94]. These data consist of 2,178 measurements on 33 nights, with two isolated epochs in April 2006 and March 2007, and the remaining data covering October 2007 to March 2008. In spite of the substantially longer baseline than TESS, a Lomb–Scargle periodogram analysis of the WASP measurements reveals no notable peaks besides the half-day, one-day and two-day sampling aliases. The lack of photometric modulation is reflected in the spectroscopic data as well; we do not detect periodic variation in the activity-sensitive Ca II H & K, Na I or Hα spectral lines as measured by the NEID DRP. Also, there is no emission in the Ca II H & K line cores in the NEID and HARPS-N spectra, suggesting that the star is chromospherically quiet.

## Stellar characterization of TIC 241249532

TIC 241249530 shares a common parallax and proper motion with TIC 241249532 as measured by Gaia, and the two stars are separated on the sky by 4.930 ± 0.104″ (ref. 86). The probability of a chance alignment between TIC 241249530 and TIC 241249532 is $R$ = 9.73 × 10$^{-5}$ (ref. 95), suggesting that the pair is indeed gravitationally bound. Gaia's photometric measurements of TIC 241249532 place it firmly along the main sequence. We do not perform an independent SED analysis on this star but instead estimate its mass using empirical mass–luminosity relations[96,97]. We calculate the mass to be 0.453 ± 0.012 times that of the Sun based on the 2MASS $K_s$-band magnitude and 0.400 ± 0.016 times that of the Sun based on the Gaia $G_{RP}$-band magnitude. The stellar mass, coordinates and broadband photometry are given in Extended Data Table 1.

On the basis of the weighted mean of the Gaia parallax measurements for the system, the on-sky separation corresponds to a projected physical separation of 1,664.00 ± 10.85 AU. The relative motions of these two stars are not constrained well enough by Gaia to meaningfully estimate an orbital solution. However, as an effort to quantify the dynamical impact of TIC 241249532 in our analysis, we simulated 10 million orbits sampled randomly in phase, uniformly in cos$i$ and thermally ($f(e) = 2e$) in eccentricity. We determine the orbital period of the system to be >10,000 years, with a peak in frequency at 35,000 years. Long-period stellar companions such as this can directly bias RV analyses of exoplanets in the form of a linear RV slope. However, our simulations show that TIC 241249532 probably induces a linear trend in the observed RVs of TIC 241249530 at the level of just 1 cm s$^{-1}$ year$^{-1}$, with 99% of our orbits returning slopes <30 cm s$^{-1}$ year$^{-1}$. The amplitude of this signal is small compared with the km-s$^{-1}$-level variations induced by TIC 241249530 b and we therefore do not include it as an extra body in our joint fitting.

## Joint transit + RV analysis

We use the exoplanet software package[98] to fit a transit model and a Keplerian orbit with Rossiter–McLaughlin perturbations to the observed photometric and RV signals for TIC 241249530. The exoplanet

package relies on starry[99,100] and the underlying analytic models from ref. 101 to fit the transits, and the orbital parameter posteriors are sampled using the PyMC3 Hamiltonian Monte Carlo package[102]. The orbit model consists of a full Keplerian with tight Gaussian priors on the orbital period, $P$, and time of conjunction, $T_0$, broad uniform priors on the exoplanet mass, $M_p$, transit impact parameter, $b$, and transit depth, $\delta$, and Gaussian priors on the stellar mass, $M_\star$, and radius, $R_\star$. We reparameterize the eccentricity, $e$, and argument of periastron, $\omega$, as $\sqrt{e}\sin\omega$ and $\sqrt{e}\cos\omega$ and we sample these on the unit disk. We do not impose an extra eccentricity prior, as the global warm-Jupiter eccentricity distribution is not well constrained[103]. Separate quadratic limb-darkening coefficients, reparameterized as $q_1$ and $q_2$ as in ref. 104, are used for each instrument for which in-transit observations were taken (TESS, ARCTIC, NEID). For the RV data, we fit individual zero-point offset terms ($\sigma$) and jitter terms ($\gamma$) for each instrument, splitting the NEID RVs into two separate datasets before and after the instrument restart. Dilution terms are included for both TESS and ARCTIC, as both transit measurements suffered from flux contamination. The TESS data products already account for dilution, but previous works have demonstrated that these results are susceptible to overcorrection[105,106], so we allow the TESS dilution term to float uniformly from 0.1 to 1.2. For ARCTIC, we fix the dilution to be 0.9947 based on the out-of-transit data for which the target was well resolved from its companion. To model the Rossiter–McLaughlin signal, we adopt the formalism of ref. 107 along with their prior distributions for the Gaussian line dispersion parameter, $\beta$, the Lorentzian line dispersion parameter, $\gamma$, and macroturbulence, $\zeta$. We place a Gaussian prior on the projected stellar rotational velocity, $v\sin i_\star$, and a uniform prior on the projected spin–orbit misalignment, $\lambda$. The prior distributions and posterior results for all of these fit parameters, as well as for some derived values, are given in Extended Data Table 2.

## Stellar obliquity

The stellar obliquity, $\psi$, is related to the projected obliquity, $\lambda$, by

$$\cos\psi = \sin i_\star \sin i \cos\lambda + \cos i_\star \cos i.$$

Here $i_\star$ is the inclination of the stellar spin axis and $i$ is the inclination of the exoplanet orbit. We cannot directly calculate $\psi$ because the stellar inclination is not known. Instead, assuming that the stellar inclination is drawn from an isotropic distribution, uniform in $\cos i_\star$, we use the above equation to determine the possible values of $\psi$ and their relative probabilities. For our derived posteriors on $\lambda$ and $i$, we find that the orbit is indeed retrograde (that is, $\psi > 90°$) at 99.5% confidence in this scenario, and we calculate the obliquity to be $\psi = 141^{+15°}_{-24}$. This value is consistent with expectations for vZLK-driven migration; simulations show that the final obliquity can be as large as 180° for systems such as this[7]. This is not definitive proof of the formation history, however, as retrograde orbits such as this can also be produced through planet–planet interactions[108,109]. Regardless, we warn that our result is strongly dependent on the naive assumption of an isotropic stellar inclination distribution, which is not always valid[110,111].

## Dynamical history—analytic constraints

The high eccentricity and tight orbit of TIC 241249530 b and the presence of the distant stellar binary companion indicate a likely history of high-eccentricity migration driven by vZLK oscillations and tidal dissipation. To determine how this formation channel could have delivered the exoplanet to its current orbit, we first identify a set of initial conditions consistent with the present-day architecture of the system. We work in the context of the secular approximation for the evolution of hierarchical triple configurations[112].

The planet is at present close enough to the primary star such that short-range forces—general relativity (GR), tides and rotational distortions—have quenched any vZLK oscillations driven by the companion. We calculate the semimajor axis at which this quenching occurs by assuming that GR dominates the short-range forces and examining the ratio between the timescale of GR precession of the inner orbit and the timescale for vZLK oscillations. To leading order (quadrupole limit), this ratio is[113]

$$\frac{t_{GR}}{t_{quad}} = \frac{a^4}{3a_2^3}\frac{(1-e^2)M_2c^2}{(1-e^2)^{3/2}G(M_\star + M_p)^2}$$

in which $M_\star$ is the mass of the primary star; $M_p$, $a$ and $e$ are the mass, semimajor axis and eccentricity of the planet; and $M_2$, $a_2$ and $e_2$ are the mass, semimajor axis and eccentricity of the binary companion. For the exoplanet and primary-star parameters, we adopt the median posteriors from our joint fit. For the binary companion, we set $M_2 = 0.453\,M_\odot$, $a_2 = 1{,}664$ AU and $e_2 = 0.5$. If the planet started with a low initial eccentricity of $e = 0.1$, vZLK oscillations would have started only if the initial semimajor axis of the planet was $a_i > 4.2$ AU, for which this value is calculated by setting $t_{GR}/t_{quad} = 1$.

We can now constrain the initial eccentricity by requiring that the periastron distance of the first vZLK oscillation was sufficiently small to trigger efficient tidal dissipation. In particular, in the quadrupole limit, the quantity $a_f \equiv a(1-e_{max}^2)$ is approximately conserved throughout the tidal migration, as the orbital angular momentum is conserved both during episodes of maximum eccentricity, as well as after vZLK oscillations have been quenched. Here $e_{max}$ indicates the maximum eccentricity reached during a vZLK oscillation and $a_f$ is equal to the final semimajor axis once the orbit has fully circularized. If $a_f$ is taken to be conserved, the maximum eccentricity of the initial vZLK oscillation must have been $e_{i,max} > 0.9947$.

Exciting an eccentricity this high on the initial vZLK oscillation must have required a substantial initial inclination, $I_i$, between the orbit of the planet and that of the binary companion. We derive a lower bound on $I_i$ using the following equation from ref. 112:

$$\varepsilon_{GR}\left(\frac{1}{j_{i,min}} - 1\right) = \frac{9}{8}\frac{e_{i,max}^2}{j_{i,min}^2}\left(j_{i,min}^2 - \frac{5}{3}\cos^2 I_i\right).$$

Here $j_{i,min} = \sqrt{1-e_{i,max}^2}$ and we have assumed GR perturbations to be dominant over those from tidal and rotational distortion. The dimensionless quantity $\varepsilon_{GR}$ measures the 'strength' of perturbations from GR relative to those of the binary companion and it is defined as

$$\varepsilon_{GR} \equiv 3G(M_\star + M_p)^2 a_2^3(1-e_2^2)^{3/2}/(M_2 a^4 c^2).$$

Extended Data Fig. 5 shows the required initial inclination between the planetary and binary orbital planes and the resulting initial maximum eccentricity with respect to the initial semimajor axis of the planet. Although vZLK oscillations are present when $a_i > 4.2$ AU, not all values above this threshold yield defined values for $I_i$ because the short-range forces are too strong for the planet to reach the required high initial eccentricity unless the initial semimajor axis exceeds $a_i > 7.0$ AU. The maximum eccentricity of the initial vZLK cycle must have been $e_{i,max} > 0.9947$ to generate the present-day semimajor axis and eccentricity. Attaining a maximum eccentricity this large is only possible with a nearly polar initial inclination between the orbit planes of the planet and the binary companion. We find that the initial inclination $I_i > 86.8°$ for $a_i > 7.0$ AU. Altogether, these results indicate that it is possible to reach the present-day parameters of the system if the planet started beyond $a_i > 7.0$ AU and the binary companion started on an orbit nearly perpendicular to that of the planet.

## Dynamical history—simulations

We now use our derived constraints on the initial orbital conditions to explore the planetary orbital evolution through numerical simulations.

We conduct integrations of the secular equations of motion for TIC 241429530 through KozaiPy, a publicly available software package that simulates hierarchical three-body systems (https://github.com/djmunoz/kozaipy). The equations of motion are provided in ref. 3. We adopt initial values of $a_i = 10$ AU and $e_i = 0.1$, considering the minimum semimajor axis necessary for vZLK oscillations to be present. We consider perturbations to the octupole order and also account for tidal evolution in the constant-time-lag model of equilibrium tides[114]. Tidal parameters are adjusted so that the system reaches its present-day orbital parameters at an age of 3 Gyr, approximately equal to the derived age of the system. Specifically, the Love number of the planet is set to $k_2 = 0.25$ and its viscous timescale is set to $t_v = 0.01$ days.

The simulation results are presented in Extended Data Fig. 6. They indicate the presence of vZLK oscillations that trigger periods of very large eccentricities. At the times that the periapse distance is minimized, tidal dissipation is strong and the semimajor axis shrinks. Eventually, the semimajor axis becomes small enough that the vZLK oscillations are suppressed owing to short-range forces and the planet decouples from the binary companion. After the vZLK oscillations are quenched, the mutual inclination is approximately conserved and the eccentricity of the planet slowly damps owing to continued tidal dissipation. We observe that there is an instant in time at which the eccentricity and semimajor axis of the planet are very close to the present-day values. We also note that the value of $a(1 - e_{max}^2)$ is conserved during episodes of maximum eccentricity of each individual vZLK cycle, ranging within only a few percent of the average value of $a(1 - e_{max}^2)$. According to this simulation, continued dissipation will cause the planet to reach a circular orbit in about a billion years. Altogether, this simulation provides a plausible proof of concept of the system's dynamical history of coupled vZLK oscillations and tidal migration. We suggest future work on the system to explore the role that dynamical tides might have played in its formation.

## Modelling the transiting-warm-Jupiter eccentricity distribution

To explore the relation between exoplanet mass and eccentricity for warm Jupiters, or intermediate-period giant planets, we start with the sample of all transiting exoplanets with masses between 0.3 and 15 times that of Jupiter and orbital periods between 10 and 365 days. For each system in this sample, we adopt the most up-to-date mass and eccentricity constraints for which the eccentricity was fit as a free parameter when solving for the orbit. We discard four planets for which all literature solutions assumed a circular orbit with the eccentricity fixed at 0. All of these planets are less than 1.3 Jupiter masses. We also remove two planets in P-type circumbinary orbits, as the dynamical environments of these systems are expected to differ from those of planets orbiting single stars[115,116]. Our sample differs from that analysed in ref. 41, which draws from the RV planet sample and thus uses projected planet mass ($M_p \sin i$) instead of true mass. By restricting our analysis to transiting exoplanets, we ensure that the measured masses are not degenerate with orbital inclination. This approach also mitigates the susceptibility of our results to detection biases, as the completeness fractions of transit surveys should be largely insensitive to exoplanet mass in the Jupiter-sized-planet regime.

The median mass of our sample is 1.935 Jupiter masses. We divide the sample into two groups of equal size, placing planets more massive than the median into one group and planets less massive than the median into the other group. The population-level eccentricity distribution of each group is then modelled in PyMC[117] using a hierarchical Bayesian framework. For our model, we adopt a beta distribution with two hyperparameters, $\theta = \{\mu, \kappa\}$, in which $\mu$ describes the mean of the distribution and $1/\kappa$ describes its variance. These hyperparameters represent a reparameterization of the standard beta distribution parameters $\alpha$ and $\beta$, in which $\alpha = \mu\kappa$ and $\beta = (1 - \mu)\kappa$. A beta distribution is chosen for its flexibility in shape and because it is naturally bounded between 0 and 1. We adopt a uniform hyperprior for $\mu \sim U(0, 1)$ and a

log-normal hyperprior for $\log \kappa \sim N(0, 3)$. These choices reduce the impact of hyperprior choices on the inference results, especially when the sample size is small, as is the case here[118]. The best-fit distributions are shown in Fig. 3 and the resulting hyperparameters are $\mu_{low} = 0.18^{+0.04}_{-0.03}$, $\log \kappa_{low} = 1.23^{+0.27}_{-0.29}$, $\mu_{high} = 0.44^{+0.05}_{-0.05}$ and $\log \kappa_{high} = 0.91^{+0.22}_{-0.24}$. The mean values, $\mu$, of the eccentricity distributions of low-mass and high-mass transiting warm Jupiters differ by $4.2\sigma$. To assess the robustness of this result, we repeated the process for mass cutoffs between 1 Jupiter mass and 2.7 Jupiter masses. These bounds were chosen such that the size ratio of the two groups does not exceed 2:1. At each cutoff, we ran 1,000 trials, drawing the planet masses from asymmetric Gaussian distributions with means and widths determined by their literature values and uncertainties. For all mass-cutoff values over this range, the mean values of the two eccentricity distributions differ by $3-5\sigma$.

## Data availability

The TESS data products referenced and analysed in this work are publicly available through the Mikulski Archive for Space Telescopes (MAST) at https://exo.mast.stsci.edu/. The raw NEID, HPF and HARPS-N spectra are available on request. Source data are provided with this paper.

## Code availability

AstroImageJ is publicly available and can be downloaded at https://www.astro.louisville.edu/software/astroimagej/installation_packages/. The exoplanet software package is open source and public and can be downloaded at https://github.com/exoplanet-dev/exoplanet. The PyMC software package is open source and public and can be downloaded at https://github.com/pymc-devs/pymc.

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

**Acknowledgements** A.F.G. thanks B. Dawson and J. Najita for guidance. NEID is financed by the National Aeronautics and Space Administration (NASA) through the Jet Propulsion Laboratory (JPL) by contract 1547612 and the NEID Data Reduction Pipeline is financed through JPL contract 1644767. Funding for this work was partially provided by Research Support Agreements 1646897 and 1679618 administered by JPL. The Center for Exoplanets and Habitable Worlds and the Penn State Extraterrestrial Intelligence Center are supported by the Pennsylvania State University and the Eberly College of Science. This research has made use of the SIMBAD database, operated at Strasbourg astronomical Data Center (CDS), Strasbourg, France, and NASA's Astrophysics Data System Bibliographic Services. This research was carried out, in part, at JPL, California Institute of Technology, under a contract with NASA (80NM0018D0004). Computations for this research were performed on the Pennsylvania State University's Institute for Computational and Data Sciences Advanced Cyberinfrastructure (ICDS-ACI). This content is solely the responsibility of the authors and does not necessarily represent the views of the Institute for Computational and Data Sciences. This paper contains data taken with the NEID instrument, which was financed by the NASA-NSF Exoplanet Observational Research (NN-EXPLORE) partnership and built by Pennsylvania State University. NEID is installed on the WIYN telescope, which is operated by the National Science Foundation (NSF)'s National Optical-Infrared Astronomy Research Laboratory, and the NEID archive is operated by the NASA Exoplanet Science Institute at the California Institute of Technology. Some of the observations in this paper made use of the NN-EXPLORE Exoplanet Stellar Speckle Imager (NESSI). NESSI was financed by the NASA Exoplanet Exploration Program and the NASA Ames Research Center. NESSI was built at the Ames Research Center by S. B. Howell, N. Scott, E. P. Horch and E. Quigley. NN-EXPLORE is managed by JPL, California Institute of Technology under contract with NASA. This work includes data collected by the TESS mission, which are publicly available from the Mikulski Archive for Space Telescopes (MAST). Funding for the TESS mission is provided by the NASA Science Mission Directorate. We acknowledge the use of public TESS data from pipelines at the TESS Science Office and at the TESS Science Processing Operations Center. Resources supporting this work were provided by the NASA High-End Computing (HEC) Program through the NASA Advanced Supercomputing (NAS) Division at Ames Research Center for the production of the SPOC data products. This research has made use of the Exoplanet Follow-up Observing Program website, which is operated by the California Institute of Technology, under contract with NASA under the Exoplanet Exploration Program. Some of the data presented in this paper were obtained from MAST. Support for MAST for non-HST data is provided by the NASA Office of Space Science through grant NNX09AF08G and by other grants and contracts. This work has made use of data from the European Space Agency (ESA) mission Gaia, processed by the Gaia Data Processing and Analysis Consortium (DPAC). Funding for the DPAC has been provided by national institutions, in particular the institutions participating in the Gaia Multilateral Agreement. This research has made use of the NASA Exoplanet Archive, which is operated by the California Institute of Technology, under contract with NASA under the Exoplanet Exploration Program. This research was made possible through the use of the AAVSO Photometric All-Sky Survey (APASS), financed by the Robert Martin Ayers Sciences Fund and NSF AST-1412587. We thank N. Leroux for assistance with the Unistellar observations. C.I.C. acknowledges support by NASA Headquarters through an appointment to the NASA Postdoctoral Program at Goddard Space Flight Center, administered by ORAU through a contract with NASA. W.D.C. acknowledges financial support from the NSF through grant AST-2108801. E.P. and I.C. acknowledge financial support from the Agencia Estatal de Investigación of the Ministerio de Ciencia e Innovación MCIN/AEI/10.13039/501100011033 and the ERDF 'A way of making Europe' through project PID2021-125627OB-C32 and from the Centre of Excellence 'Severo Ochoa' award to the Instituto de Astrofísica de Canarias. C.X.H. thanks the support of the ARC DECRA project DE200101840. G.N. acknowledges research funding from the Ministry of Education and Science programme the 'Excellence Initiative – Research University' conducted at the Centre of Excellence in Astrophysics and Astrochemistry of the Nicolaus Copernicus University in Toruń, Poland. T.M. acknowledges financial support from the Spanish Ministry of Science and Innovation (MICINN) through the Spanish State Research Agency, under the Severo Ochoa Program 2020–2023 (CEX2019-000920-S). D.D. acknowledges support from the NASA Exoplanet Research Program grant 18-2XRP18_2-0136

and from the TESS Guest Investigator Program grant 80NSSC23K0769. G. Stefánsson acknowledges support provided by NASA through the NASA Hubble Fellowship grant HST-HF2-51519.001-A awarded by the Space Telescope Science Institute, which is operated by the Association of Universities for Research in Astronomy, Inc., for NASA, under contract NAS5-26555. Based in part on observations at Kitt Peak National Observatory, NSF's NOIRLab (proposal IDs 2021A-0388, 2021B-0442, 2022A-627532, 2022B-176691, 2023A-383728, 2023A-722344, 2023B-607177 and 2024A-843425; PI: A. Gupta), managed by the Association of Universities for Research in Astronomy (AURA) under a cooperative agreement with the NSF. We thank the WIYN Observing Associates for their support of our NEID observations. We are honoured to be permitted to conduct astronomical research on Iolkam Du'ag (Kitt Peak), a mountain with particular importance to the Tohono O'odham. We also thank Z. Arnold, J. Davis, M. Edwards, J. Ehret, T. Juan, B. Pisarek, A. Rowe, F. Wortman, the Eastern Area Incident Management Team and all of the firefighters and air support crew who fought the recent Contreras Fire and saved Kitt Peak National Observatory. These results are based on observations obtained with the Habitable-zone Planet Finder Spectrograph on the Hobby–Eberly Telescope (HET). The HPF team acknowledges support from NSF grants AST-1006676, AST-1126413, AST-1310885, AST-1517592, AST-1310875, ATI 2009889, ATI-2009982 and AST-2108512 as well as the NASA Astrobiology Institute (NNA09DA76A) in the pursuit of precision radial velocities in the near-infrared. The HPF team also acknowledges support from the Heising-Simons Foundation through grant 2017-0494. The HET is a joint project of the University of Texas at Austin, the Pennsylvania State University, Ludwig-Maximilians-Universitaet Muenchen and Georg-August Universität Göttingen. The HET is named in honour of its principal benefactors, W. P. Hobby and R. E. Eberly. The Texas Advanced Computing Center (TACC) at the University of Texas at Austin provided high-performance computing, visualization and storage resources that have contributed to the results reported in this paper. C.I.C. and S.V. are NASA Postdoctoral Fellows. B.G., R.K., L.L., M.L., M.P., M.S., G. Simard and S.W. are Unistellar Citizen Scientists. G. Stefánsson is a NASA Sagan Fellow.

**Author contributions** A.F.G. selected TIC 241249530 for ground-based observations, designed and led the NEID, NESSI and HPF observing programmes, performed the transit and radial velocity analysis, coordinated the analysis of the transiting-warm-Jupiter mass–eccentricity distribution and wrote most of the manuscript. S.C.M. and H.I. performed the analysis of the dynamical history and trajectory of TIC 241249530b and contributed the associated text and figures. J.D. wrote the code to analyse the eccentricity distributions and contributed to the theoretical interpretation of the results. J.M.J. contributed to the theoretical interpretation of the eccentricity distribution results. I.C. helped coordinate the HARPS-N observations. J.L.-R., M.D. and S.M. coordinated and conducted the ARCTIC observations and data processing. M.R.G. contributed to the characterization of TIC 241249532. A.S.J.L. performed the SERVAL analysis of the NEID spectra and S.K. performed the SERVAL analysis of the HPF spectra. X-Y.W. performed the iSpec analysis of the NEID spectra. D.D. coordinated the efforts of the TESS Single Transit Planet Candidate Working Group. K.S. performed the SED analysis. D.J.S. ran an independent analysis of the stellar SED to validate the results. T.M. performed the BACCHUS analysis of the HARPS-N spectra. S.M. and J. Wright contributed to the interpretation of the significance of the TIC 241249530 system in the context of the exoplanet population and to the coordination of follow-up observations. M.E.E. conducted the NESSI observations, generated the reconstructed speckle images and calculated the contrast limits. S.E.L., H.S., E.G., J.H., J.K., P.F. and Y.P. scheduled and executed the NEID observations. J.A.A.-M., C.B., C.H.B., C.I.C., S.H., S.K., A.S.J.L., S.M., M.W.M., A.M., J.P.N., J.R., P.R., A.R., C.S., G. Stefánsson and J. Wright contributed to the design, development and commissioning of the NEID spectrograph and data-reduction software. C.B., W.D.C., S.K., J.P.N., P.R., A.R. and G. Stefánsson contributed to the design, development and commissioning of the HPF spectrograph and data-reduction software. D.P.S. assisted with the coordination of the HPF observations. G.N. and E.P. coordinated and conducted the HARPS-N observations. L.H. contributed to the development and installation of the diffuser used to obtain the ARCTIC observations. L.H. and J. Wisniewski helped coordinate and execute the ARCTIC observations. S.V. developed the pipeline with which the TESS transit signal for TIC 241249530b was first detected and P.D., J.P. and C.Z. vetted the alerted signals in TESS Sector 20 and helped validate the transit of TIC 241249530b. C.X.H. developed the Quick Look Pipeline and D.C. developed the TESS-SPOC pipeline. B.G., R.K., L.L., M.L., M.P., M.S. and G. Simard conducted the Unistellar observations and L.S. analysed the Unistellar data.

**Competing interests** The authors declare no competing interests.

**Additional information**
**Correspondence and requests for materials** should be addressed to Arvind F. Gupta.

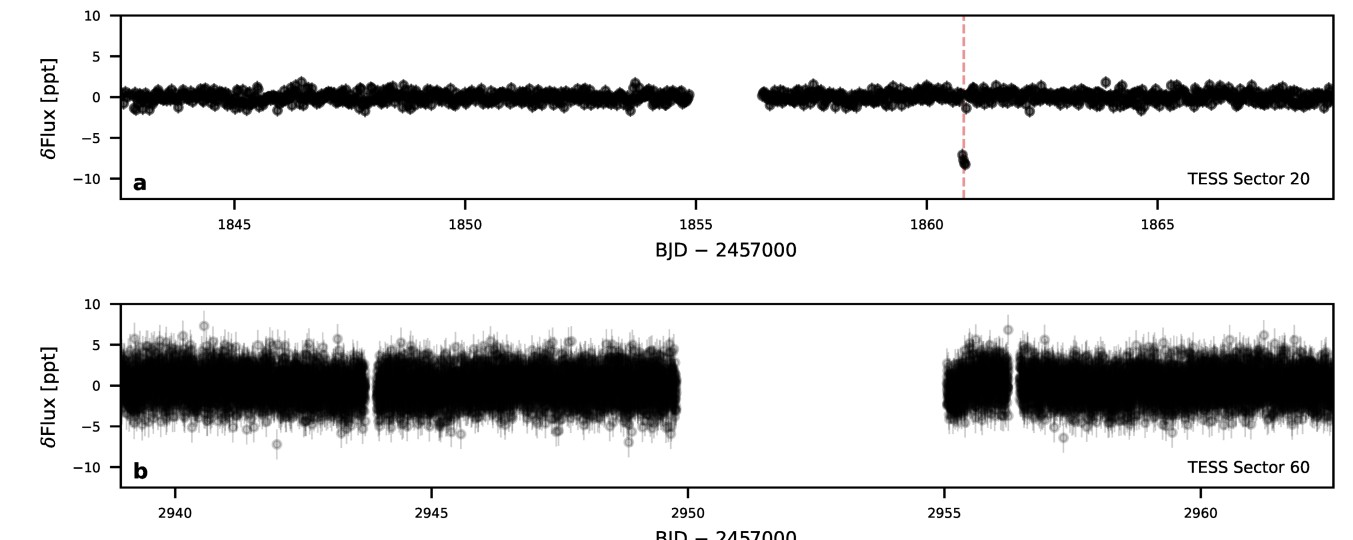

**Extended Data Fig.1 | TESS light curves for TIC 241249530. a**, Sector 20 TESS-SPOC data (30-min cadence). The vertical dashed line marks the best-fit transit midpoint. **b**, Sector 60 SPOC data (2-min cadence). Error bars on individual points represent 1σ measurement uncertainties.

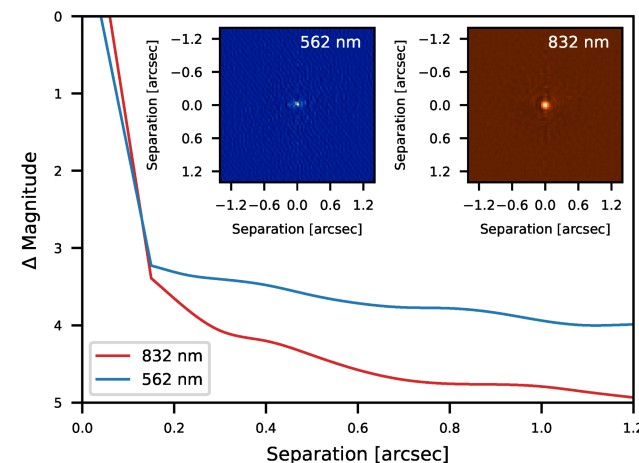

**Extended Data Fig. 2 | Reconstructed NESSI speckle images and 5σ contrast curves for TIC 241249530.** Observations were taken simultaneously at 562 nm with the blue camera (upper-left inset image) and at 832 nm with the red camera (upper-right inset image). The contrast curves indicate the limiting magnitude difference at which bound or background sources could be detected for separations between 0.2″ and 1.2″.

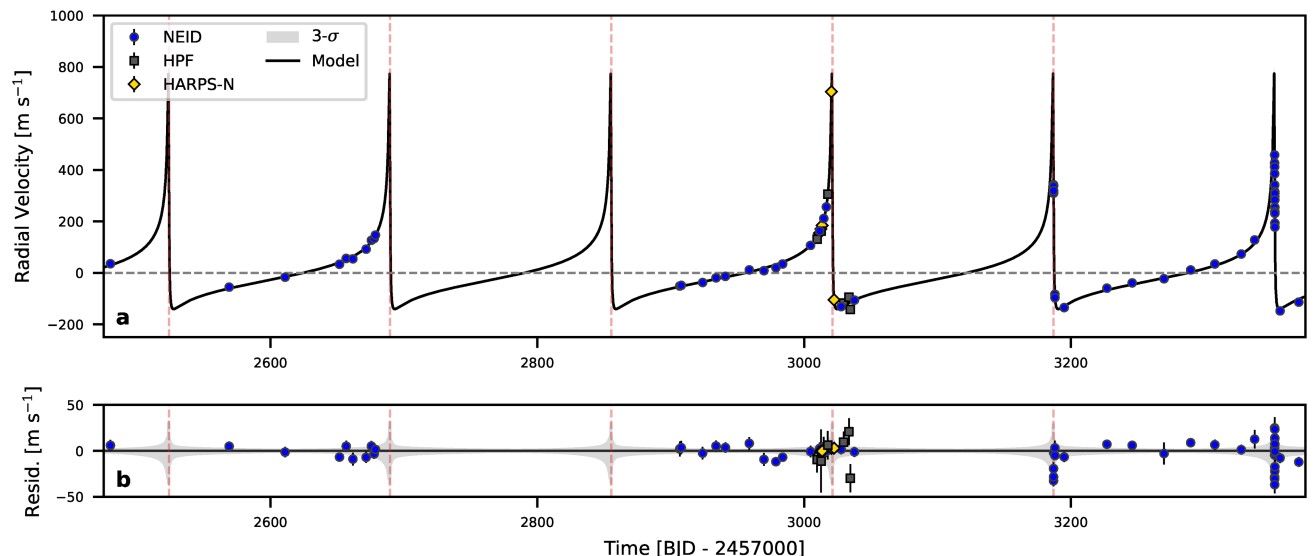

**Extended Data Fig. 3 | RV time series for TIC 241249530. a**, RV measurements from NEID (blue), HPF (black) and HARPS-N (gold) and best-fit orbit model (black curve). **b**, Residuals to the RV orbit fit. Vertical red lines in both panels mark the predicted transit times. The grey-shaded region bounds the $3\sigma$ confidence intervals for the fit. The horizontal axis is in units of days relative to Barycentric Julian Date (BJD) 2457000. Error bars on individual points represent $1\sigma$ measurement uncertainties.

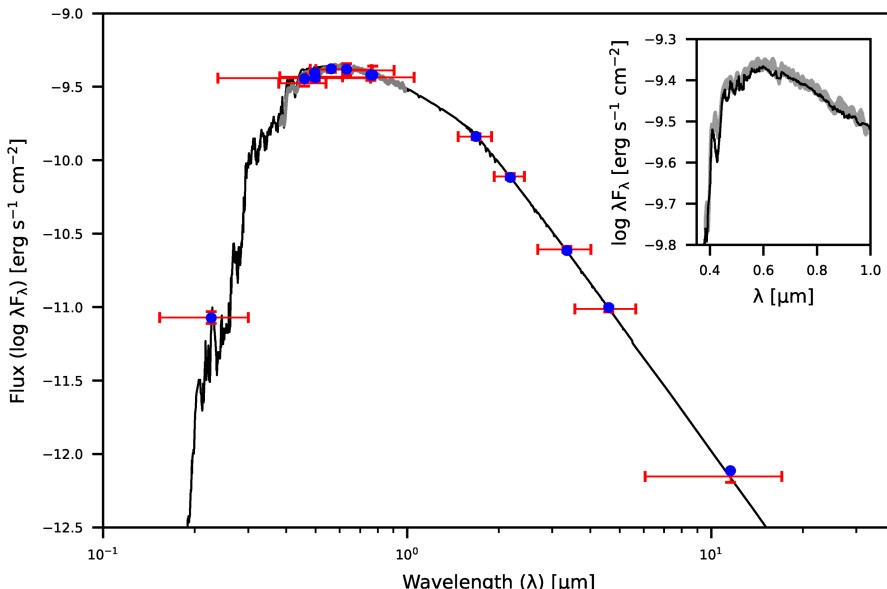

**Extended Data Fig. 4 | SED for TIC 241249530.** Red symbols represent the observed photometric measurements, for which the vertical error bars represent the 1σ measurement uncertainties and the horizontal bars represent the effective width of the passband. Blue symbols are the model fluxes from the best-fit PHOENIX atmosphere model (black). The Gaia spectrophotometry is represented as a grey swathe (see also inset plot).

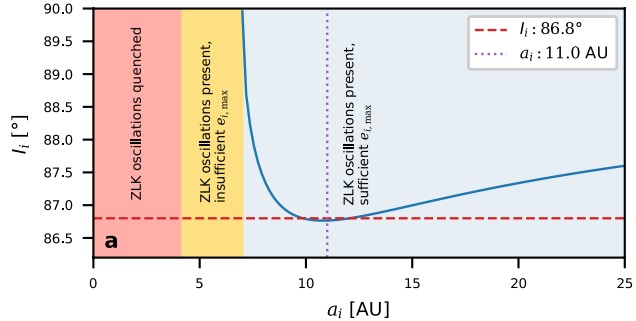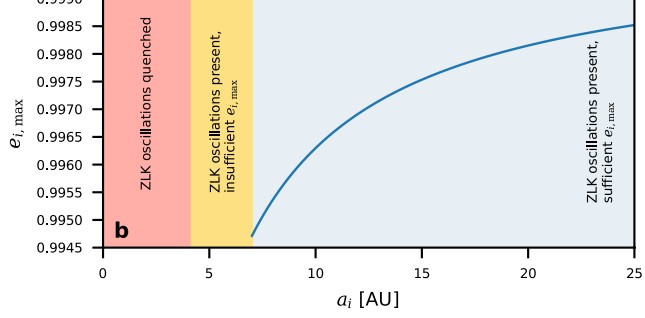

**Extended Data Fig. 5 | Constraints on the parameters of the first vZLK oscillation for TIC 241249530 b. a**, Constraints on initial inclination. **b**, Constrains on initial maximum eccentricity. The red region (0 AU < $a_i$ < 4.2 AU) indicates the absence of vZLK oscillations. The orange region (4.2 AU < $a_i$ < 7 AU) indicates the presence of vZLK oscillations but with insufficient $e_{i,max}$ to reach the present-day orbit owing to strong short-range forces (primarily GR, with further contributions from tides and rotational distortions). The blue region ($a_i$ > 7 AU) indicates the presence of vZLK oscillations that can drive the planet to reach present-day parameters.

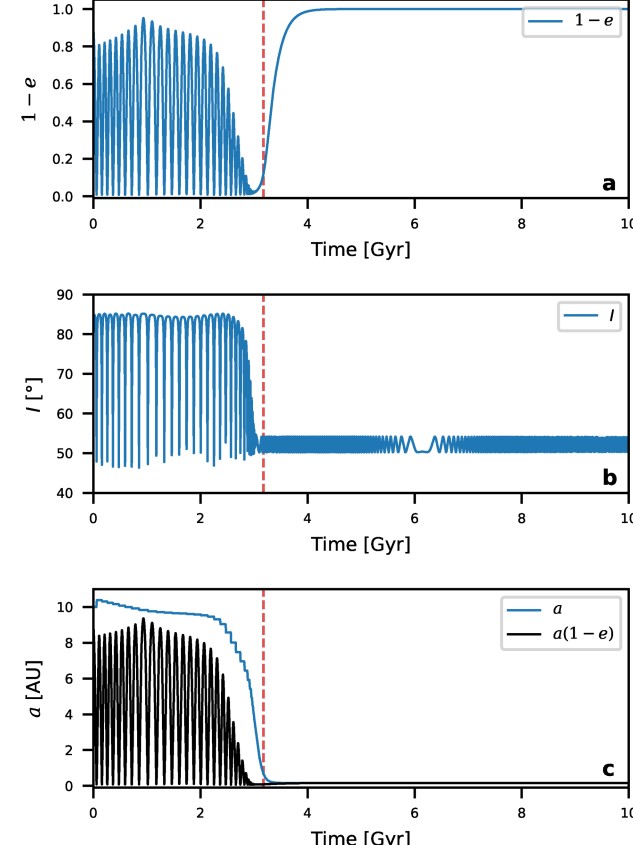

**Extended Data Fig. 6 | Simulated evolution of the orbit of TIC 241249530 b resulting from high-eccentricity migration driven by vZLK oscillations.** **a**, Evolution of the eccentricity of the planetary orbit over time. **b**, Evolution of the mutual inclination between the planet and binary orbits over time. **c**, Evolution of the semimajor axis, $a$, and periastron separation, $a(1 − e)$, of the planetary orbit over time. The vertical red lines at 3.2 Gyr mark the age at which the orbit reaches the present-day conditions ($e = 0.94$, $a = 0.64$ AU). We adopt $a_i = 10$ AU, $e_i = 0.1$ for the initial orbital parameters for illustrative purposes.

**Extended Data Table 1 | Stellar parameters for TIC 241249530 and TIC 241249532**

| Parameter | Value | | Unit | Ref. |
|---|---|---|---|---|
| Identifiers: | | | | |
| TIC | 241249530 | 241249532 | | 119 |
| *Gaia* DR3 | 995754258812449024 | 995754254517548032 | | 86 |
| 2MASS | J06211411+5317423 | J06211389+5317380 | | 84 |
| Coordinates and Parallax: | | | | |
| R.A. | 06:21:14.02 | 06:21:13.76 | J2000 | 86 |
| Dec | +53:17:42.32 | +53:17:37.85 | J2000 | 86 |
| Parallax | $2.96 \pm 0.02$ | $3.17 \pm 0.09$ | mas | 86 |
| Proper Motion R.A. | $7.620 \pm 0.018$ | $7.766 \pm 0.107$ | mas yr$^{-1}$ | 86 |
| Proper Motion Dec | $-15.717 \pm 0.015$ | $-15.799 \pm 0.092$ | mas yr$^{-1}$ | 86 |
| Broadband photometry: | | | | |
| $B$ | $12.226 \pm 0.413$ | | mag | 87 |
| $V$ | $11.689 \pm 0.039$ | | mag | 87 |
| TESS | $11.1328 \pm 0.0079$ | $16.5809 \pm 0.0314$ | mag | 119 |
| $G$ | $11.5593 \pm 0.0003$ | $17.4606 \pm 0.0029$ | mag | 86 |
| $B_p$ | $11.8835 \pm 0.0007$ | $18.046 \pm 0.090$ | mag | 86 |
| $R_p$ | $11.0698 \pm 0.0005$ | $16.054 \pm 0.026$ | mag | 86 |
| $J$ | 10.538 | 12.017 | mag | 84 |
| $H$ | $10.266 \pm 0.024$ | $13.741 \pm 0.064$ | mag | 84 |
| $K_s$ | $10.192 \pm 0.021$ | $13.506 \pm 0.071$ | mag | 84 |
| $W_1$ | $10.115 \pm 0.023$ | | mag | 85 |
| $W_2$ | $10.147 \pm 0.021$ | | mag | 85 |
| $W_3$ | $10.176 \pm 0.065$ | | mag | 85 |
| $W_4$ | 8.993 | | mag | 85 |
| Derived Stellar Parameters: | | | | |
| $T_{eff}$ | $6166 \pm 43$ | | K | iSpec |
| | $6150 \pm 50$ | | K | BACCHUS |
| log $g$ | $4.23 \pm 0.06$ | | log (cm s$^{-2}$) | iSpec |
| | $4.25 \pm 0.10$ | | log (cm s$^{-2}$) | BACCHUS |
| [Fe/H] | $0.09 \pm 0.03$ | | dex | iSpec |
| | $0.20 \pm 0.09$ | | dex | BACCHUS |
| $v \sin i_\star$ | $4.2 \pm 0.8$ | | km s$^{-1}$ | iSpec |
| | $5.5 \pm 0.7$ | | km s$^{-1}$ | BACCHUS |
| $M_\star$ | $1.24 \pm 0.07$ | $0.453 \pm 0.012$ | M$_\odot$ | |
| $R_\star$ | $1.404 \pm 0.028$ | | R$_\odot$ | |
| $L_{\star,bol}$ | $2.56 \pm 0.08$ | | L$_\odot$ | |
| $F_{\star,bol}$ | $7.19 \pm 0.20 \times 10^{-10}$ | | erg s$^{-1}$ cm$^{-2}$ | |
| Age | $3.2 \pm 0.5$ | | Gyr | |
| $A_V$ | $0.31 \pm 0.02$ | | mag | |

Data from refs. 84–87,119.

## Extended Data Table 2 | Joint-fit priors and posteriors

| Parameter | Prior | Posterior | Unit |
|---|---|---|---|
| **Stellar Parameters:** | | | |
| $M_\star$ | $\mathcal{N}(1.24, 0.07)$ | $1.271^{+0.061}_{-0.068}$ | $M_\odot$ |
| $R_\star$ | $\mathcal{N}(1.404, 0.028)$ | $1.397^{+0.025}_{-0.028}$ | $R_\odot$ |
| $v \sin i_\star$ | $\mathcal{N}(4.2, 0.8)$ | $4.60^{+0.56}_{-0.63}$ | km s$^{-1}$ |
| $\zeta$ | $\mathcal{U}(2.0, 6.5)$ | $3.6^{+1.7}_{-1.0}$ | km s$^{-1}$ |
| $\beta$ | $\mathcal{U}(0.8, 1.2)$ | $1.01^{+0.13}_{-0.12}$ | km s$^{-1}$ |
| $\gamma$ | $\mathcal{U}(2.5, 4.5)$ | $3.37^{+0.60}_{-0.54}$ | km s$^{-1}$ |
| $q_{1,\text{TESS}}$ | $\mathcal{U}(0.0, 1.0)$ | $0.35^{+0.29}_{-0.19}$ | |
| $q_{2,\text{TESS}}$ | $\mathcal{U}(0.0, 1.0)$ | $0.17^{+0.29}_{-0.26}$ | |
| $q_{1,\text{ARCTIC}}$ | $\mathcal{U}(0.0, 1.0)$ | $0.47^{+0.28}_{-0.23}$ | |
| $q_{2,\text{ARCTIC}}$ | $\mathcal{U}(0.0, 1.0)$ | $0.14^{+0.40}_{-0.35}$ | |
| $q_{1,\text{NEID}}$ | $\mathcal{U}(0.0, 1.0)$ | $0.48^{+0.39}_{-0.28}$ | |
| $q_{2,\text{NEID}}$ | $\mathcal{U}(0.0, 1.0)$ | $-0.01^{+0.32}_{-0.26}$ | |
| **Orbital and Planetary Parameters:** | | | |
| $T_0$ | $\mathcal{N}(2458860.8, 0.01)$ | $2458860.8007^{+0.0015}_{-0.0016}$ | BJD |
| $P$ | $\mathcal{N}(165.77, 0.1)$ | $165.77190^{+0.00027}_{-0.00028}$ | days |
| $\sqrt{e}\cos\omega$ | $\mathcal{U}(-1.0, 1.0)$ | $0.7174^{+0.0042}_{-0.0046}$ | |
| $\sqrt{e}\sin\omega$ | $\mathcal{U}(-1.0, 1.0)$ | $0.6531^{+0.0050}_{-0.0046}$ | |
| $e$ | derived | $0.9412^{+0.0009}_{-0.0009}$ | |
| $\omega$ | derived | $42.32^{+0.40}_{-0.36}$ | deg |
| $b$ | $\mathcal{U}(0.0, 1.0)$ | $0.581^{+0.049}_{-0.058}$ | |
| $a$ | derived | $0.641^{+0.010}_{-0.012}$ | AU |
| $\lambda$ | $\mathcal{U}(-180, 180)$ | $163.5^{+9.4}_{-7.7}$ | deg |
| $R_p/R_\star$ | $\mathcal{U}(0.05, 0.15)$ | $0.0873^{+0.0019}_{-0.0020}$ | |
| $M_p$ | $\mathcal{U}(0.5, 30)$ | $4.98^{+0.16}_{-0.18}$ | $M_J$ |
| $R_p$ | derived | $1.186^{+0.037}_{-0.040}$ | $R_J$ |
| **Instrument Parameters:** | | | |
| TESS Dilution | $\mathcal{U}(0.1, 1.2)$ | $1.010^{+0.069}_{-0.067}$ | |
| $\sigma_{\text{phot,TESS}}$ | $\mathcal{L}(0.74, 10)$ | $0.004^{+0.010}_{-0.004}$ | ppt |
| $\sigma_{\text{phot,ARCTIC}}$ | $\mathcal{L}(4.18, 10)$ | $1.51^{+0.11}_{-0.10}$ | ppt |
| $\sigma_{\text{RV,NEID-pre}}$ | $\mathcal{U}(0, 25)$ | $3.1^{+2.6}_{-1.9}$ | m s$^{-1}$ |
| $\sigma_{\text{RV,NEID-post}}$ | $\mathcal{U}(0, 25)$ | $4.6^{+1.6}_{-1.7}$ | m s$^{-1}$ |
| $\sigma_{\text{RV,HPF}}$ | $\mathcal{U}(0, 25)$ | $14.6^{+7.1}_{-8.4}$ | m s$^{-1}$ |
| $\sigma_{\text{RV,HARPS-N}}$ | $\mathcal{U}(0, 25)$ | $3.3^{+3.9}_{-2.1}$ | m s$^{-1}$ |
| $\gamma_{\text{RV,NEID-pre}}$ | $\mathcal{N}(0, 1000)$ | $-57.2^{+1.6}_{-1.6}$ | m s$^{-1}$ |
| $\gamma_{\text{RV,NEID-post}}$ | $\mathcal{N}(0, 1000)$ | $-70.1^{+1.3}_{-1.3}$ | m s$^{-1}$ |
| $\gamma_{\text{RV,HPF}}$ | $\mathcal{N}(0, 1000)$ | $-18.9^{+3.9}_{-4.1}$ | m s$^{-1}$ |
| $\gamma_{\text{RV,HARPS-N}}$ | $\mathcal{N}(0, 1000)$ | $-170.7^{+3.1}_{-2.7}$ | m s$^{-1}$ |

We report the median values of the posterior distributions from our joint fit. Uncertainties represent the 68% confidence intervals (±1σ) for each parameter. Limb darkening is sampled using the parameterization given by ref. 104. For λ, we use the custom angle distribution from the PyMC3 extras extension of the exoplanet package[98] to avoid discontinuities at ±π.