## [Peer Review file · Nature]

Manuscript Title: A hot Jupiter progenitor on a super-eccentric, retrograde orbit

Reviewer Comments & Author Rebuttals

Reviewer Reports on the Initial Version:

Referees' comments:

Referee #1 (Remarks to the Author):

This paper presents an analysis of the photometric and radial velocity (RV) measurements of the transiting planet TIC 241249530b. This reveals a 5 MJup mass planet on an eccentric 165-d orbit. This is indeed a very interesting system in that it qualifies as the most eccentric transiting Jupiter rivaled only by HD 80606b. TIC 241249530b may thus help in our understanding how planets in eccentric orbits form. Adding to the interest of this exoplanet is the fact that the authors have measured the Rossiter-McLaughlin (R-M) effect and this indicates that the planet is in a retrograde orbit.

The data are superb and it is a beautiful orbit (Fig. 2). Normally, I like to do my own analysis on the RV data to confirm the orbit, but there is not much point in this case. The orbit is very convincing. The authors suggest that this high eccentricity stems from the Kozai (vZLK) mechanism and back their claim with numerical simulations. They wisely claim that this is "plausible" evidence for vZLK causing the high eccentricity. One may never know with certainty until the orbit of the stellar binary is known. This is impossible (for now) given the $> 10,000$ yr orbital period. They also propose that vZLK may be source for the high eccentric massive giant planets. My guess is that there are several mechanisms at play (planet-planet interaction, disk-planet interaction, etc.) creating the mass-eccentricity distribution.

Overall I found this to be a very good paper: Well written, excellent quality data, sound analysis with reasonable conclusions. The fact that this is one of only two such systems merits publication in Nature. I recommend publication after the authors consider my relatively minor comments:

1) In Fig. 3 the authors plot the mass-eccentricity for warm transiting Jupiters. Why restrict it to just transiting exoplanets? If you make this figure for all giant planets in the mass range 0.3 - 10 MJup, regardless of whether they are transiting or not (i.e. Jupiter that are not necessarily "warm") you find that these all nicely follow the distribution of points in the diagram. How will this affect your conclusions? Probably most of the host stars will not have binary companions so the vZLK may not come into play.

2) The authors refer to these as "warm" Jupiters. But "warm" is a very subjective term and different people may have different definitions. The authors should be more clear by stating an orbital distance they consider giant planet to be "warm". (Say, for solar-like stars since this of course depends on the mass of the host star.) In the modeling of the distribution in "Methods" the authors considered planets at orbital periods of 10 - 365 d which would be in the "hot" to "temperate" range

for Jupiters (my definition). One should give more details by replacing vague adjectives with numbers.

3) line 88 in the introduction: "..migrated inward from further away"

Again a case of using of a vague expression to replace a detailed description. What is meant by "further away"? A planet at 0.1 AU is "further way" from a planet at 0.05 AU, yet they both migrated in via the same mechanism. The consensus is that these giant planets migrated from where they formed somewhere beyond the ice line at ~ 3 AU. "Further away" really does not convey to the reader from where these planets originated.

4) I was a bit surprised that that authors present intriguing evidence that the planet is in a retrograde orbit, yet they only have scant discussion on this. I realize that there are only four data points and that more measurements are needed, but it really looks like the in-transit RV measurements show the inverse R-M effect (i.e. retrograde orbit). If true, how would that affect your conclusions? To me it seems that the vZLK mechanism cannot produce a planet in a retrograde orbit. It can only change the orbital eccentricity and inclination, not the spin axis. One mechanism that can do this is planet-planet interactions. This could throw the planet in an eccentric *and* retrograde orbit. Doesn't this go counter to the claim that the eccentricity is due to vZLK? To me it is an important aspect of this exoplanet which can provide an important clue to the formation, yet there is little

Referee #2 (Remarks to the Author):

The authors have discovered and characterized a transiting planet, TIC 241249530 b, which breaks the record for the most eccentric planet known. Its mass is rather higher than the typical Jupiter-class planets, and this is a property shared with the planet with now the second-highest eccentricity, HD 80606 b. The authors notice that both planetary systems also have a companion star beyond 1000AU, which can generate the high eccentricities by the vZLK dynamical mechanism, as long as they subsequently experienced tidal dissipation (which is ongoing). Less massive planets may not be able to survive that same journey. This inspired the authors to analyze all the transiting Jupiter-class planets, and they find the higher-mass planets follow a distribution that is generally more eccentric orbits than the lower-mass planets.

The detection of the new planet is certainly both novel and interesting. It secures the vZKL theory as a process really happening in nature, with a misaligned planet and a stellar companion acting as smoking guns. This had already been achieved with HD80606b, but it is interesting to start building a population because then details of the dissipation process might have observational guidance. I was curious about whether the less secure vZKL migrators, quoted in references 26-29, support this general picture.

Other points in the paper were only modest additions to the literature:

- 1) The detection of an eccentricity dependence on planetary mass is not novel. I believe it was first commented on by Butler, Wright, Marcy et al 2006, ApJ, 646, 505. It was considered as a constraint on planet-planet scattering models by Juric & Tremaine 2008, ApJ, 686, 603 and claimed as a success for the model of Ford & Rasio 2008, ApJ, 686, 621. It was shown again by Wright et al. 2009, ApJ, 693, 1084. The current demonstration builds on these by specializing to transiting planets, i.e. those with better-determined masses, but since it is a statistical signal that aspect was not crucial.
- 2) The tidal dissipation mechanism for making hot Jupiters was already known to depend on planetary mass, having been introduced by Gu, Bodenheimer, and Lin 2003, ApJ, 588, 5090, and modeled in this very context by Rozner, Glanz, Perets, & Grishin 2022, ApJ, 931, 10. My understanding of the literature is that the low-mass giant planets are not necessarily destroyed, but they can simply become circular while losing some mass: a rocky core can halt runaway disruption (Liu, Guillochon, Lin & Ramirez-Ruiz 2013, ApJ, 762, 37). It would pull them out of the high-eccentricity population fast, so it can contribute to the observational effect. So I am saying the big-picture theory in this paper is not novel, but the new observational constraint of a “population” of vZLK-migrators is novel.

The quality of the analysis was high, the data are clear, and the presentation is excellent.

One aspect I felt may be less secure than it seems is the shape of the Rossiter effect. Since there is no out-of-transit RV data very near the Rossiter-constraining ones, the orbital model here might be rather uncertain. Over 48 hours, the HARPS-N RV changes by -808 m/s, i.e. at -17 m/s/hr spanning the time of transit (it is likely somewhat more negative than that at the time of transit; the model is most uncertain here, but in addition to RVs it is constrained by the transit phase). The NEID data during transit change by -23 m/s/hr, which seems about like the orbital rate. But figure 2c has the orbital effect subtracted, and it is surprising to me that the Rossiter model still rises by +30 m/s/hr.

I'm saying naively, it seems like less of an effect would be detected. And I am pretty certain the Rossiter parameters should be degenerate with the orbital parameters until a more complete spectroscopic dataset is obtained both inside and just outside of transit.

Other aspects of statistics were done well, including the demonstration of a difference in eccentricity between higher- and lower-mass gas giants. I liked the theoretical constraints on the original configuration of the planet; the simulation was more than what is necessary (as it is done many times in the literature), but it is nice to see it applied to this particular system.

The conclusion about low-mass planets being disrupted during vZLK cycles is not secure: they can be circularized quickly. Also, the conclusion about the origin of the eccentricity dependence on mass is not secure: the paragraph from 217-232 suggests there are plenty of other ways to explain these trends. Reasoning from the specific to the general, in lines 230-232, is not logically valid.

I think the title and abstract are great, i.e. I do not think they press this conclusion inappropriately.

A minor precision issue: the separation on line 135 should be written as 1664 ± 11 AU.

I do not have suggestions for additional experiments needed for this paper. But I would like to see some of the literature referenced above cited.

Daniel Fabrycky

Author Rebuttals to Initial Comments:

Referees' comments:

Referee #1 (Remarks to the Author):

This paper presents an analysis of the photometric and radial velocity (RV) measurements of the transiting planet TIC 241249530b. This reveals a 5 MJup mass planet on an eccentric 165-d orbit. This is indeed a very interesting system in that it qualifies as the most eccentric transiting Jupiter rivaled only by HD 80606b. TIC 241249530b may thus help in our understanding how planets in eccentric orbits form. Adding to the interest of this exoplanet is the fact that the authors have measured the Rossiter-McLaughlin (R-M) effect and this indicates that the planet is in a retrograde orbit.

The data are superb and it is a beautiful orbit (Fig. 2). Normally, I like to do my own analysis on the RV data to confirm the orbit, but there is not much point in this case. The orbit is very convincing. The authors suggest that this high eccentricity stems from the Kozai (vZLK) mechanism and back their claim with numerical simulations. They wisely claim that this is "plausible" evidence for vZLK causing the high eccentricity. One may never know with certainty until the orbit of the stellar binary is known. This is impossible (for now) given the $> 10,000$ yr orbital period. They also propose that vZLK may be source for the high eccentric massive giant planets. My guess is that there are several mechanisms at play (planet-planet interaction, disk-planet interaction, etc.) creating the mass-eccentricity distribution.

Overall I found this to be a very good paper: Well written, excellent quality data, sound analysis with reasonable conclusions. The fact that this is one of only two such systems merits publication in Nature. I recommend publication after the authors consider my relatively minor comments:

1) In Fig. 3 the authors plot the mass-eccentricity for warm transiting Jupiters. Why restrict it to just transiting exoplanets? If you make this figure for all giant planets in the mass range $0.3 - 10 M_{\text{Jup}}$, regardless of whether they are transiting or not (i.e. Jupiter that are not necessarily "warm") you find that these all nicely follow the distribution of points in the diagram. How will this affect your conclusions? Probably most of the host stars will not have binary companions so the vZLK may not come into play.

Below we show Fig. 3 with the addition of all planets in the mass range $0.3 M_J < M \sin i < 10 M_J$ and measured eccentricities (data from the NASA Exoplanet Archive accessed on 9 April 2024). As the referee suggests, these do appear to follow the same distribution. This result is expected given that the mass-eccentricity trend has been discussed in the literature, as Referee #2 notes, and it is also seen in the RV-only sample of Freilikh et al. 2019, ApJ, 884, 57 (reference #30 in our manuscript).

The fact that this trend exists outside of our narrowly defined population suggests that the mass dependence of the eccentricity distribution is not inherently tied to the hot or warm Jupiter formation process, as we stated in our initial draft. So while we still ascribe some importance to the fact that the trend exists within the transiting warm Jupiter population, we have softened our language to make it clear that it does not clearly point to vZLK or any single formation mechanism.

2) The authors refer to these as "warm" Jupiters. But "warm" is a very subjective term and different people may have different definitions. The authors should be more clear by stating an orbital distance they consider giant planet to be "warm". (Say, for solar-like stars since this of course depends on the mass of the host star.) In the modeling of the distribution in "Methods" the authors considered planets at orbital periods of 10 - 365 d which would be in the "hot" to "temperate" range for Jupiters (my definition). One should give more details by replacing vague adjectives with numbers.

We have replaced most instances of the term "warm Jupiters" with "intermediate-period giant planets", and we provide a quantitative definition ($10 \text{ d} < P < 365 \text{ d}$; $0.3 \text{ MJ} < M < 15 \text{ MJ}$) as the referee suggests. Still, we do keep the more qualitative term "warm Jupiters" in some instances to avoid excess verbosity, with the surrounding context making it clear that we are referring to the aforementioned population.

3) line 88 in the introduction: "...migrated inward from further away"

Again a case of using of a vague expression to replace a detailed description. What is meant by "further away"? A planet at 0.1 AU is "further way" from a planet at 0.05 AU, yet they both migrated in via the same mechanism. The consensus is that these giant planets migrated from where they formed somewhere beyond the ice line at ~ 3 AU. "Further away" really does not convey to the reader from where these planets originated.

We have rephrased this as “migrated in from beyond the ice line” to make it clear we are talking about planets that formed like Jupiter and Saturn.

4) I was a bit surprised that that authors present intriguing evidence that the planet is in a retrograde orbit, yet they only have scant discussion on this. I realize that there are only four data points and that more measurements are needed, but it really looks like the in-transit RV measurements show the inverse R-M effect (i.e. retrograde orbit). If true, how would that affect your conclusions? To me it seems that the vZLK mechanism cannot produce a planet in a retrograde orbit. It can only change the orbital eccentricity and inclination, not the spin axis. One mechanism that can do this is planet-planet interactions. This could throw the planet in an eccentric *and* retrograde orbit. Doesn't this go counter to the claim that the eccentricity is due to vZLK? To me it is an important aspect of this exoplanet which can provide an important clue to the formation, yet there is little

The vZLK mechanism can produce planets on retrograde orbits. This result is seen consistently in the simulations performed by Anderson, Storch, & Lai 2016, MNRAS, 456, 3671 (reference #7 in our manuscript), who also discuss the process in depth. This does require expanding the gravitational potential of the binary companion to the octupole order, which can in some cases can lead to tidal disruption (Petrovich 2015, ApJ, 799, 27; reference #6 in our manuscript), but the Anderson et al. results show that it can indeed produce a retrograde hot Jupiter or warm Jupiter population. We also note that our own simulations of vZLK-driven migration can naturally produce retrograde orbits; the following plot shows the obliquity evolution for one such simulation with initial conditions that also reproduce the present observed orbit.

Ψ (Stellar spin-orbit misalignment) vs time

Referee #2 (Remarks to the Author):

The authors have discovered and characterized a transiting planet, TIC 241249530 b, which breaks the record for the most eccentric planet known. Its mass is rather higher than the typical Jupiter-class planets, and this is a property shared with the planet with now the second-highest eccentricity, HD 80606 b. The authors notice that both planetary systems also have a companion star beyond 1000AU, which can generate the high eccentricities by the ν ZLK dynamical mechanism, as long as they subsequently experienced tidal dissipation (which is ongoing). Less massive planets may not be able to survive that same journey. This inspired the authors to analyze all the transiting Jupiter-class planets, and they find the higher-mass planets follow a distribution that is generally more eccentric orbits than the lower-mass planets.

The detection of the new planet is certainly both novel and interesting. It secures the ν ZKL theory as a process really happening in nature, with a misaligned planet and a stellar companion acting as smoking guns. This had already been achieved with HD80606b, but it is interesting to start building a population because then details of the dissipation process might have observational guidance. I was curious about whether the less secure ν ZKL migrators, quoted in references 26-29, support this general picture.

All three of these planets have $e \sim 0.7$. HD 17156 b ($M = 3.24 M_J$) and TOI-3362 b ($M = 4 M_J$) have high masses, but KOI-1257 b ($M = 1.45 M_J$) falls in the low-mass population.

Other points in the paper were only modest additions to the literature:

1) The detection of an eccentricity dependence on planetary mass is not novel. I believe it was first commented on by Butler, Wright, Marcy et al 2006, ApJ, 646, 505. It was considered as a constraint on planet-planet scattering models by Juric & Tremaine 2008, ApJ, 686, 603 and claimed as a success for the model of Ford & Rasio 2008, ApJ, 686, 621. It was shown again by Wright et al. 2009, ApJ, 693, 1084. The current demonstration builds on these by specializing to transiting planets, i.e. those with better-determined masses, but since it is a statistical signal that aspect was not crucial.

Thank you for pointing us to these previous works. We have restructured the discussion to make it clear that our identification of a mass-eccentricity correlation is not novel, and we add references to several of these papers. We instead focus on the finding that transiting warm Jupiters conform to this known trend, which is not something that has previously been discussed in the literature.

2) The tidal dissipation mechanism for making hot Jupiters was already known to depend on planetary mass, having been introduced by Gu, Bodenheimer, and Lin 2003, ApJ, 588, 5090, and modeled in this very context by Rozner, Glanz, Perets, & Grishin 2022, ApJ, 931, 10. My understanding of the literature is that the low-mass giant planets are not necessarily destroyed, but they can simply become circular while losing some mass: a rocky core can halt runaway disruption (Liu, Guillochon, Lin & Ramirez-Ruiz 2013, ApJ, 762, 37). It would pull them out of the high-eccentricity population fast, so it can contribute to the observational effect. So I am saying the big-picture theory in this paper is not novel, but the new observational constraint of a “population” of vZLK-migrators is novel.

Again, thank you for highlighting these related papers. We have added the Rozner reference and revised the text to hopefully add some clarity about the outcomes of tidal disruption, but we were unable to add any thorough discussion given the length constraints of the manuscript.

The quality of the analysis was high, the data are clear, and the presentation is excellent.

One aspect I felt may be less secure than it seems is the shape of the Rossiter effect. Since there is no out-of-transit RV data very near the Rossiter-constraining ones, the orbital model here might be rather uncertain. Over 48 hours, the HARPS-N RV changes by -808 m/s, i.e. at -17 m/s/hr spanning the time of transit (it is likely somewhat more negative than that at the time of transit; the model is most uncertain here, but in addition to RVs it is constrained by the transit phase). The NEID data during transit change by -23 m/s/hr, which seems about like the orbital rate. But figure 2c has the orbital effect subtracted, and it is surprising to me that the Rossiter model still rises by +30 m/s/hr. I'm saying naively, it seems like less of an effect would be detected. And I am pretty certain the Rossiter parameters should be degenerate with the orbital parameters until a more complete spectroscopic dataset is obtained both inside and just outside of transit.

We acknowledge that the lack of out-of-transit measurements in the original data set could be cause for concern. We obtained a new measurement of the Rossiter-McLaughlin effect on 12 February 2024, while the paper was under review. This consists of a full in-transit RV baseline as well as several RV measurements before ingress and after egress. We re-fit the orbit using these new data as well as all previous data. All of the fitted parameters are consistent with the original values to better than 1-sigma, but we note that the posteriors for those parameters most closely affected by the Rossiter-McLaughlin signal are now much tighter, as expected.

Other aspects of statistics were done well, including the demonstration of a difference in eccentricity between higher- and lower-mass gas giants. I liked the theoretical constraints on the original configuration of the planet; the simulation was more than what is necessary (as it is done many times in the literature), but it is nice to see it applied to this particular system.

The conclusion about low-mass planets being disrupted during ν ZLK cycles is not secure: they can be circularized quickly. Also, the conclusion about the origin of the eccentricity dependence on mass is not secure: the paragraph from 217-232 suggests there are plenty of other ways to explain these trends. Reasoning from the specific to the general, in lines 230-232, is not logically valid.

We agree that the line of reasoning was not consistent, so we have removed the conclusion about the origin of the eccentricity dependence. We instead remain agnostic about this origin, and suggest that multiple mechanisms can be responsible.

I think the title and abstract are great, i.e. I do not think they press this conclusion inappropriately.

A minor precision issue: the separation on line 135 should be written as 1664 ± 11 AU.

Thank you, this has been corrected.

I do not have suggestions for additional experiments needed for this paper. But I would like to see some of the literature referenced above cited.